# Satellite evidence of substantial rain-induced soil emissions of ammonia across the Sahel

Jonathan E. Hickman[1, *], Enrico Dammers[2], Corinne Galy-Lacaux[3], Guido R. van der Werf[1]

[1]Earth and Climate Cluster, Vrije Universiteit, Amsterdam, 1081 HV, Netherlands

[2]Environment and Climate Change Canada, Toronto, Ontario, M3H 5T4, Canada

[3]Laboratoire d'Aérologie UPS-CNRS UMR 5560, Toulouse, 31400, France

*now at NASA Goddard Institute for Space Studies, New York, NY, 10025, USA

*Correspondence to*: Jonathan E. Hickman (jonathan.e.hickman@nasa.gov)

**Abstract.** Atmospheric ammonia ($NH_3$) is a precursor to fine particulate matter formation and contributes to nitrogen deposition, with potential implications for the health of humans and ecosystems. Agricultural soils and animal excreta are the primary source of atmospheric $NH_3$, but natural soils can also be an important emitter. In regions with distinct dry and wet seasons such as the Sahel, the start of the rainy season triggers a pulse of biogeochemical activity in surface soils known as the Birch effect, which is often accompanied by emissions of microbially-produced gases such as carbon dioxide and nitric oxide. Field and lab studies have sometimes, but not always, observed pulses of $NH_3$ after the wetting of dry soils; however, the potential regional importance of these emissions remains poorly constrained. Here we use satellite retrievals of atmospheric $NH_3$ using the Infrared Atmospheric Sounding Interferometer (IASI) regridded at 0.25° resolution, in combination with satellite-based observations of precipitation, surface soil moisture, and nitrogen dioxide concentrations, to present evidence of substantial precipitation-induced pulses of $NH_3$ across the Sahel at the onset of the rainy season in 2008. The highest concentrations of $NH_3$ occur in pulses during March and April, when $NH_3$ biomass burning emissions estimated for the region are low. For the region of the Sahel spanning 10° to 16° N and 0° to 30° E, changes in $NH_3$ concentrations are weakly but significantly correlated with changes in soil moisture during the period from mid-March through April, when the peak $NH_3$ concentrations occur ($r=0.28$, $p=0.02$). The correlation is also present when evaluated on an individual pixel-basis during April ($r=0.16$,

p<0.001).  Using a simple box model, average emissions for the entire Sahel are between 2 and 6 mg $NH_3$ $m^{-2}$ $day^{-1}$ during peaks of the observed pulses, depending on the assumed effective lifetime. These early season pulses are consistent with surface observations of monthly deposition, which show an uptick in $NH_3$ deposition at the start of the rainy season for sites in the Sahel. The $NH_3$ concentrations in April are also correlated with increasing tropospheric $NO_2$ concentrations observed by the Ozone Monitoring Instrument (r=0.78, p<0.0001), which have previously been attributed to the Birch effect.  Box model results suggest that pulses occurring over a 35-day period in March and April are responsible for roughly one fifth of annual $NH_3$ emissions from the Sahel. We conclude that precipitation early in the rainy season is responsible for substantial $NH_3$ emissions in the Sahel, likely representing the largest instantaneous fluxes of nitrogen gas from the region during the year.

## 1. Introduction

Ammonia ($NH_3$) plays an important role in the atmosphere and in the nitrogen (N) cycle.  In the atmosphere, $NH_3$ is a precursor to the formation of fine particular matter ($PM_{2.5}$), which contributes to substantial levels of premature mortality (Lelieveld *et al*., 2015).  $NH_3$ can also form a substantial proportion of atmospheric N deposition (Dentener *et al*., 2006; Holland *et al*., 2005), affecting downwind ecosystems by potentially altering productivity (Thomas *et al*., 2009), soil pH (Tian and Niu, 2015), eutrophication status (Bergstrom and Jannson, 2006), biodiversity (Bobbink *et al*., 2010), and stimulating emissions of other trace gases such as nitric oxide (NO) and nitrous oxide ($N_2O$; e.g., Eickenscheidt *et al*., 2011; Pilegaard *et al*., 2006).

Cropland and grazed soils have long been known to be a major source of atmospheric ammonia through the volatilization of urea and ammonium ($NH_4^+$) based inorganic fertilizers as well as of livestock excreta (Bouwman *et al*., 1997). Ammonia emissions can also represent an important N flux in natural ecosystems, particularly in drylands. In deserts, soil $NH_3$ emissions can represent over 25% of annual nitrogen losses (McCalley and Sparks, 2008) and over 10% in a semiarid savanna (Fiona M Soper, 2016).

Soil moisture is a key control over biogeochemical cycles in drylands (Austin *et al.*, 2004). Biogeochemical cycling in dryland soils is often characterized by pulsing dynamics related to the wetting or re-wetting of dry soils, known as the Birch effect (Birch, 1960; Birch and Friend, 1956).  In environments where the distribution of annual precipitation is distinctly seasonal, soil microbial activity typically declines during the dry season, as water becomes limiting and microbes senesce or become dormant (Borken and Matzner, 2009).  N may build up in soils during this period, when little biological uptake occurs, but when atmospheric N deposition continues and senesced microbial and plant material accumulates (Borken and Matzner, 2009).  The onset of the rainy season can initiate a rapid increase in microbial activity (Birch and Friend, 1956; Borken and Matzner, 2009; Placella and Firestone, 2013; Placella *et al.*, 2012), as re-awakened microbes take advantage of pools of N that accumulated or were made more bioavailable during the dry season, leading to large increases in N mineralization rates (Birch, 1958; 1960; Borken and Matzner, 2009; Dijkstra *et al.*, 2012; Saetre and Stark, 2004; Semb and Robinson, 1969).

The abrupt change in water potential also represents a stress to microbes, prompting a flush of labile N solutes released by microbes to maintain turgor pressure (Kieft *et al.*, 1987). This increase in microbial activity is accompanied by pulsed emissions of trace gases such as carbon dioxide ($CO_2$; Emmerich, 2003; Huxman *et al.*, 2004; Saetre and Stark, 2004) and NO (e.g., Anderson and Levine, 1986; Davidson, 1992), which can be an important component of annual emissions (Davidson, 1992; Jaeglé *et al.*, 2004).  As the availability of $NH_4^+$ is a major control over $NH_3$ volatilization (Nelson, 1982; Schlesinger and Peterjohn, 1991), a flush of N mineralization and an increase in soil moisture would be expected to trigger an $NH_3$ emission pulse as well. A few studies have documented pulse dynamics in emissions of $NH_3$ in laboratory or field settings (Delon *et al.*, 2017; McCalley and Sparks, 2008; Schlesinger and Peterjohn, 1991; Soper *et al.*, 2016). In arid and semi-arid ecosystems, soil $NH_3$ emissions have been observed to increase by ~15% to 630% after wetting (Schlesinger and Peterjohn, 1991). However, there are few studies outside desert ecosystems (Kim *et al.*, 2012), and an increase in $NH_3$ emissions following wetting is not always observed (e.g., Yahdjian and Sala, 2010).  The potential importance of these pulse emissions of $NH_3$ at landscape or regional scales remains poorly constrained.

In addition to $NH_4^+$ availability, soil pH is a key environmental control over $NH_3$ production in soils (Dawson, 1977; Nelson, 1982). Since $NH_3$ is typically produced through the deprotonation of $NH_4^+$, $NH_3$ emissions would be expected to be higher in relatively alkaline soils, or in soil with alkaline

microsites. Globally, soils tend to shift from alkaline to acidic when mean annual precipitation exceeds mean annual evapotranspiration (Slessarev *et al.*, 2016), so soils in drier biomes such as deserts and grasslands tend to be alkaline (though there are exceptions to this pattern), creating conditions favourable to $NH_3$ volatilization. For example, the combination of pH and ammonium concentrations in soils from a semi-arid ecosystem in Senegal have been shown to create conditions favourable to the emission of $NH_3$

(Delon *et al.*, 2017).

Given the importance of rainfall seasonality, soil pH, and N availability in contributing to $NH_3$ emission pulses, soils in the Sahel may be an important source of $NH_3$ to the atmosphere during the onset of the rainy season, and a case study for determining whether Birch effect $NH_3$ pulsing is an important

process at broad regional scales. The Sahel is a grassland environment representing a transition between desert and productive savannas. It is characterized by a unimodal rainfall seasonality, with mean annual precipitation typically ranging between 100 and 600 mm $yr^{-1}$. Seasonal variation in rainfall is broadly determined by movement of the Intertropical Convergence Zone (ITCZ). Migration of the ITCZ north of the equator in the first half of the calendar year is accompanied by the onset of the rainy season and West

African Monsoon, with the first substantial rain events occurring in April. The southward retreat of the ITCZ marks the dry season in the Sahel starting in October or November. Recent maps of African soils based on surface reflectance suggest that soils across the Sahel tend to have pHs largely near neutral, but can be higher than 9 in some areas (Vågen *et al.*, 2016). The combination of seasonal rainfall variability and soils with neutral or alkaline pHs suggests that Sahelian soils may be an important source of $NH_3$ at

the onset of the rainy season. Although the Sahel has regions of relatively dense cropland, it is characterized by lower levels of fertilizer inputs (FAO, accessed 2018) and smaller loads of atmospheric N deposition (Dentener *et al.*, 2006; Galy-Lacaux and Delon, 2014; Laouali *et al.*, 2012; though deposition can be elevated at the Sahel's southern boundary) than other parts of the world. However, it

has moderately high livestock densities (Robinson *et al*., 2014), potentially providing sites of abundant available N for the production of $NH_3$. Indeed, soil $NH_3$ emissions have been shown to be higher at a site in northern Senegal following a rain event (Delon *et al*., 2017).

5  Earlier work using total column observations from the GOME instrument presented evidence of high atmospheric concentrations of nitrogen dioxide ($NO_2$) over the Sahel in the early rainy season of 2000, which appeared to broadly correspond to rainfall events in the region (Jaeglé *et al*., 2004). These early growing season increases in tropospheric $NO_2$ concentrations could not be attributed to lightning or to biomass burning, leaving soil emissions the presumed source. Soils emit NO through a variety of biotic

10 and abiotic mechanisms; in the atmosphere, NO rapidly interconverts to $NO_2$, and the two gases are collectively referred to as $NO_x$. Inverse modelling subsequently suggested that soil emissions of NO across sub-Saharan Africa were of the same magnitude as emissions from biomass burning, which previously had been thought to be the dominant $NO_x$ source in the region. Note that in this paper, NO is used in discussions of soil emissions specifically. Since satellite observations are of $NO_2$, we use $NO_2$

15 when discussing those observations, and $NO_2$ or $NO_x$ when discussing modelled surface emissions based on those observations.

  At regional scales, the processes controlling seasonal variability in emissions, and the magnitude of emission responses to these controls, are not well constrained. Atmospheric models often rely on static

20 emissions inventories of $NH_3$ that lack intra-annual variability or detailed environmental controls over emissions from soils (e.g., Bouwman *et al*., 1997; European Commission, Joint Research Center (JRC)/Netherlands Environmental Agency (PBL), 2011; Lamarque *et al*., 2010), particularly for natural ecosystems (e.g., Paulot and Jacob, 2014). Fire emission inventories are generally created using data on burned area, fuel load, combustion completeness to compute dry matter of carbon losses, and emission

25 factors that translate these into trace gas or aerosol emissions, producing daily emissions estimates at 0.25° resolution (e.g., van der Werf *et al*., 2017). Inventories of $NH_3$ emissions from natural soils tend to rely on a global estimate for the year 1990 (Bouwman *et al*., 1997) and agricultural emission inventories

with sub-annual temporal resolution tend not to consider environmental controls other than temperature and wind speed (e.g., Paulot and Jacob, 2014).

Here we use satellite retrievals of atmospheric $NH_3$ concentrations (Whitburn *et al*., 2016) over Africa to evaluate whether the onset of the rainy season causes pulsed emissions of $NH_3$ over the Sahel, focusing on the year 2008, and evaluate its environmental drivers. We compare the seasonal pattern in atmospheric $NH_3$ concentrations observed by satellite to monthly surface $NH_3$ deposition measured at 7 sites in north equatorial Africa. We also use a simple box model to calculate surface fluxes based on retrieved atmospheric concentrations, and compare modelled surface fluxes to $NH_3$ emissions from biomass burning as quantified in the Global Fire Emissions Database 4s (GFED4s; van der Werf *et al*., 2017). Finally, we also make comparisons to modelled surface fluxes of $NO_2$ derived from $NO_2$ observations made by the Ozone Monitoring Instrument (OMI; Krotkov, accessed 2018).

## 2. Methods

### 2.1 Satellite products

The Infrared Atmospheric Sounding Interferometer (IASI-A), launched aboard the European Space Agency's MetOp-A in 2006, obtains retrievals of atmospheric $NH_3$ at a global distribution and bi-daily resolution. IASI-A is a polar-orbiting instrument in a sun-synchronous orbit (9:30 Local Solar Time equator crossing, descending node), providing two observations daily; here we use morning observations, when the thermal contrast is more favourable for retrievals (Clarisse *et al*., 2009; Van Damme *et al*., 2014). IASI-A provides a horizontal resolution of 12 km over a swath width of about 2,200 km. The retrieval product used follows the approach implemented by Whitburn *et al.* (2016), in which total columns of $NH_3$ are obtained by calculating a dimensionless spectral index (HRI), which is then converted into a $NH_3$ total column through the use of a neural network. The neural network uses a range of variables such as temperature and water vapor profiles to represent the state of the atmosphere as best as possible to produce the matching $NH_3$ total column for that atmospheric state. Retrievals with errors above 100% were excluded from the analysis, though exceptions were made for low concentrations, which tend to

have a higher error, when the following condition was met:

Retrieval error × total column $NH_3$ × 0.01 < 5 ×10$^{15}$ molecules cm$^{-2}$ (1)

where the retrieval error is a percentage and the total column $NH_3$ is a concentration in molecules cm$^{-2}$. In addition, only retrievals that were at least 75% cloud-free were used. Given the absence of hourly observations in the Sahel, the detection limit of IASI is difficult to determine with certainty. However, the region experiences high thermal contrast, and IASI seems to be able to reliably observe down to 1 to 2 ppb at the surface (is there a reference for this statement?). We regridded the Level-2 IASI $NH_3$ product to 0.25° × 0.25° resolution to match the resolution of soil moisture and other data used in the analysis. Specifically, we calculated the concentration for a given grid cell as the mean of all elliptical IASI footprints for which the corners of the grid cell were within the footprint. The IASI product has been validated using ground-based Fourier transform infrared (FTIR) observations of $NH_3$ total columns, with robust correlations at sites with high $NH_3$ concentrations, but lower at sites where atmospheric concentrations approach IASI's detection limits (Dammers *et al.*, 2017). Compared to the FTIR observations the IASI total columns are biased low by ~30% which varies per region depending on the local concentrations.

We used the Tropical Rainfall Measuring Mission (TRMM) daily precipitation product (3B42), which is based on a combination of TRMM observations, geo-synchronous infrared observations, and rain gauge observations (Huffman *et al.*, 2007). Independent rain gauge observations from West Africa have been used to validate the product, with no indication of bias in the product (Nicholson *et al.*, 2003).

We used the European Space Agency's Climate Change Initiative (ESA-CCI) 30-year daily soil moisture product gridded at 0.25° × 0.25° resolution (Dorigo *et al.*, 2017; Gruber *et al.*, 2017; Liu *et al.*, 2012). The product is based on both passive and active microwave sensors, and has been validated globally (Dorigo et al., 2014) and in East Africa (McNally *et al.*, 2016). Although the product exhibited

moderate correlation with ground observations globally, it provides relatively high correlations (r~0.7) for West African sites when seasonality is included in the analysis (Dorigo *et al.*, 2014).

We also used the publicly available level 3 tropospheric $NO_2$ concentrations product from OMI, a nadir-viewing spectrometer measuring solar backscatter in the UV-visible range aboard NASA's Aura satellite (Krotkov *et al.*, 2017, Krotkov 2018). The product is cloud-screened, including only pixels that are at least 70% cloud-free, and provided at $0.25° \times 0.25°$ resolution. The OMI product relies on air mass factors calculated with the assistance of an atmospheric chemical transport model, and is sensitive to model representations of emission, chemistry and transport data. These are generally poorly constrained for regions not commonly analysed in chemical transport models such as sub-Saharan Africa (McLinden *et al.*, 2014). Additional bias may be introduced due to the reliance on nearly cloud-free pixels, where greater sunlight may induce higher photochemical rates. For example, the current product is biased roughly 30% low over the Canadian oil sands (McLinden *et al.*, 2014). Level 2 OMI-$NO_2$ product has been validated against in situ and surface-based observations showing good agreement (Lamsal *et al.*, 2014). In statistical analyses, soil moisture, precipitation, and $NO_2$ data were masked to match pixels for which $NH_3$ retrievals were also available.

## 2.2 Surface flux calculations

A range of possible surface fluxes of $NH_3$ and $NO_2$ from our focal study region in the Sahel, ranging from 10°N to 16°N and from 0°E to 30°E, were calculated from IASI total column $NH_3$ concentrations and OMI-$NO_2$ tropospheric concentrations using a simple box model (Jacob, 1999). The specific region was selected as representative of the Sahel, and to allow for direct comparisons to earlier work examining $NO_2$ emissions from the region (Jaeglé *et al.*, 2004). Daily mean gridded concentrations of retrievals for each gas were averaged across the focal region with units of molecules $cm^{-2}$. The mean total column concentrations of gas *x* were converted to a mean surface density for the region in units of kg $m^{-2}$ for each day, using the following equation:

$$M_{x,t} = (TC_{x,t} \times MM_x \times 10)/N_a \tag{2}$$

where $M_{x,t}$ is the mean surface density of gas $x$ for day $t$, $TC_{x,t}$ is the average of retrieved total columns of gas $x$ in units of molecules cm$^{-2}$ for day $t$, $MM_x$ is the molar mass of gas $x$, and $N_a$ is Avogadro's number. The dividend on the right hand of the equation is multiplied by 10 to convert the mean surface density to units of kg m$^{-2}$. This mean surface density was then used in the box model to calculate a mean surface flux for each day, assuming first order losses of gas $x$:

$$E_{x,t}(mol) = \frac{(M_{x,t} - M_{x,t-d}e^{-d/\tau_x})}{\tau_x(1 - e^{-d/\tau_x})} \qquad (3)$$

where $E_{x,t}$ is the surface flux on day t in units of kg m$^{-2}$ day$^{-1}$, $d$ is the time between subsequent observations (one day in this study), and $\tau_x$ is the effective lifetime of a molecule of gas $x$, including both reactive and transport losses. As both the effective lifetime and the surface flux are unknowns in the equation, we use a range of plausible lifetimes for each gas to calculate a range of surface fluxes. For NH$_3$, we use lifetimes of 6, 12, 24, and 36 hours (Dentener and Crutzen, 1994; Whitburn et al., 2015); for NO$_2$, we use lifetimes of 6, 12, and 24 hours (Beirle et al., 2011; de Foy et al., 2015; Jena et al., 2014). The values were selected to reflect the possible range of lifetimes throughout the year, including periods of elevated wet deposition during the rainy season. Our box modelled NO$_2$ emissions used only satellite observations for grid cells where NH$_3$ observations were also present; model results restricted to using satellite observations for grid cells where both NH$_3$ and NO$_2$ are presented in the supplemental information.

## 2.3 Emissions inventory

GFED4s (van der Werf et al., 2017) provides monthly fire emissions at 0.25° resolution based on satellite-derived burned area (Giglio et al., 2013; Randerson et al., 2012) and a modified version of the Carnegie-Ames-Stanford-Approach (CASA) biogeochemical model (Potter et al., 1993). Daily emissions are calculated using data on the fraction of monthly emissions emitted on each date. Uncertainty in GFED fire emissions stems from uncertainty in burned area, fuel consumption, and emission factors but is poorly

constrained. According to van der Werf *et al.* (2017) a 1σ of about 50% for fire carbon emissions is reasonable for continental scale estimates. This may also be a best-guess estimate for fire $NH_3$ emissions in our study region; while the uncertainty in $NH_3$ emission factors is large because few fires have been sampled and adds to the total uncertainty, burned area and fuel consumption in savannas are, in general, better constrained than in other biomes.

## 2.4 Statistical analyses

Pearson product moment correlation analyses were conducted using pearsonr from the scipy.stats package in Python v3.6.3.

## 3. Results and Discussion

The wetting of dry soils has been known to stimulate biogeochemical cycling since at least the 1950s. Its role in creating large pulsed emissions of trace gases such as $CO_2$ and NO has been demonstrated at laboratory (e.g., Birch and Friend, 1956), field (e.g., Davidson, 1992), and regional scales through satellite observations (Jaeglé *et al.*, 2004) and observation networks (Adon *et al.*, 2010). Evidence for its importance to emissions of $NH_3$ has been limited to a few laboratory and field studies, with sometimes contrasting results, and its importance at landscape or regional scales is not well constrained. Here we present evidence that the Birch effect is an important driver of $NH_3$ emissions at regional scales, and likely responsible for the periods of the highest atmospheric $NH_3$ concentrations over the Sahel region in Africa.

### 3.1 Seasonal variability in $NH_3$ concentrations over Africa

Atmospheric $NH_3$ concentrations in 2008 exhibited broad seasonality that appears to correspond to seasonal precipitation patterns across the continent (Figure 1). In January, mean monthly concentrations are highest across a latitudinal band from roughly 5°N to 10°N, broadly corresponding to the region of highest biomass burning emissions (Figure 1). In April, higher cumulative monthly precipitation across the southern Sahel coincides with increased mean monthly $NH_3$ concentrations in the

focal region (the red box in Figure 1b); fire emissions are generally absent across the Sahel and much of the rest of the continent in April, with some emissions occurring along coastal West Africa. By August, during the middle of the West African Monsoon, hotspots of ammonia concentrations are generally absent from the continent, though concentrations are slightly elevated in central southern Africa, where emissions from biomass burning are also elevated (Figure 1).

## 3.2 Evidence for precipitation-induced emissions of $NH_3$ in the Sahel

For our focal region of the Sahel (defined above and outlined in red in Figure 1), mean atmospheric $NH_3$ concentrations exhibit two distinct peaks in late March and April (Figure 2a, highlighted in light and dark pink, respectively), which represent the highest concentrations observed in 2008. The late March peak occurs at the same time as an apparent modest increase in mean soil moisture (Fig 2b). The peak in April, during which atmospheric $NH_3$ concentrations over the Sahel are elevated relative to other parts of north equatorial Africa (Figure 3a), occurs during the first period of sustained rainfall in the focal region, and corresponds to a peak in soil moisture, suggesting a possible causal relationship between changes in soil moisture and atmospheric $NH_3$ concentrations (Figure 2b and section 3.2.1 below). This increase does occur following a possible modest increase in mean fire emissions (Fig 2a) across the Sahel. Overall, however, the seasonality in IASI-retrieved atmospheric $NH_3$ concentrations exhibits a marked difference from the seasonality in GFED4s $NH_3$ emissions from fires, which start increasing in September and peak in November (Fig 2a and section 3.2.2 below).

### 3.2.1 Soil moisture controls over early growing season atmospheric $NH_3$ concentrations

Multiple lines of evidence support a causal relationship between changes in soil moisture and atmospheric $NH_3$ concentrations across the Sahel. Foremost are the significant correlations between ESA-CCI's soil moisture product and the IASI total column during the onset of the rainy season. Specifically, for the period from February 23 to May 1, 2008, there is a significant correlation between mean atmospheric $NH_3$ concentrations of all pixels with acceptable retrievals in the Sahel focal region and mean soil moisture across the same set of pixels (r=0.28, P=0.02; Figure 4a). This correlation

integrates all pixels across the Sahel, and thus excludes any sub-regional spatial structure in the data. To examine whether a relationship between soil moisture and $NH_3$ concentrations is also present at a finer spatial scale, we conducted a correlation between the two variables for each pixel for which both variables were present during April, and again found a significant positive linear correlation (r=0.16, p<0.001; Figure 4b). Maps of 3-day averages for precipitation, soil moisture, and total column $NH_3$ concentration over the southern Sahel at 0.25° resolution illustrate the changes in each variable that occur during the development of the April emissions peak (Figure 5). Between April 13 and April 27, precipitation events in the southern half of the focal region (left-hand column) appear to be accompanied by steady increases in soil moisture (middle column) and total column $NH_3$ (right-hand column).

Although the explanatory power of the correlations between ESA-CCI's soil moisture product and the IASI total column $NH_3$ concentrations is relatively low, it demonstrates a broad regional correlation between $NH_3$ concentrations and changes in soil moisture at the onset of the rainy season – a relationship that is strong enough to be observed at the scale of the Sahel. To provide some context for the statistical result, earlier work finding that rainfall is responsible for large emissions of $NO_2$ in the same region did not include any statistical analysis in support of its findings (Jaeglé *et al*., 2004), and studies using satellite observations to infer environmental controls interpret correlation coefficients on the order of 0.25 to be evidence of a strong effect (e.g., Andela *et al*., 2017). Additionally, we would not necessarily expect a linear relationship between the quantity of soil moisture and the quantity of $NH_3$ emitted.

### 3.2.2 Minimal contribution of fire emissions to early growing season pulses

We can further exclude biomass burning as the source of the observed pulses of atmospheric $NH_3$. Biomass burning occurs in the Sahel with a well-known seasonality as based on several independent satellite fire metrics such as burned area and active fire detections (Duncan, 2003; Giglio *et al*., 2006). In general, burning in the Sahel occurs in the second half of the year, and few emissions are expected at the onset of the rainy season (Figure 1, 2). A comparison between our simple box model estimates of $NH_3$ flux and emissions from the GFED4s inventory strongly supports the hypothesis that

biomass burning does not play an important role in $NH_3$ emissions during March or April, and further suggests that biomass burning may represent a relatively unimportant regional source of $NH_3$ during most of the year, outside of the biomass burning season (Figure 6). The modelled total surface flux based on IASI observations varies depending on the effective lifetime used by up to a factor of roughly 4, with seasonal patterns mirroring the patterns in atmospheric concentration, and the largest emissions occurring during the pulses in March and April (Figure 6). In contrast, fire emissions of $NH_3$ from GFED4s are concentrated in the second half of the year, and are negligible during March and April, suggesting that they do not contribute to the early growing-season emission pulses.

The importance of a source of $NH_3$ other than biomass burning in equatorial North Africa was suggested earlier for 0 to 10° N, a region immediately south of the Sahel and of the southern boundary of our focal region in Figures 1 and 2 (Van Damme *et al.*, 2015; Whitburn *et al.*, 2015). Whitburn *et al.* observe a one to two-month lag between peak fire radiative power and IASI- $NH_3$ that differs from the pattern of IASI carbon monoxide (CO) concentrations, suggesting that fire emissions can explain only some of the atmospheric $NH_3$ present. They speculate that soils may be the source of $NH_3$ emissions following the fire season, perhaps resulting from an increase in volatilization caused by increasing soil temperatures and pH, though the paper is focused on evaluating fire emissions and they do not test this hypothesis (Whitburn *et al.*, 2015). Van Damme *et al.* speculate that the emissions are agricultural in nature (Van Damme *et al.*, 2015). IASI-$NH_3$ concentrations in the other regions examined (southern Africa, Southeast Asia, and central South America) generally track fire activity, or are thought to be due to anthropogenic emissions (Whitburn *et al.*, 2015). Although we have focused on the Sahel, it seems possible that the seasonality of rainfall in environments further south, between 0°N and 10°N, could partly explain the elevated emissions observed there following the end of the fire season.

**3.2.3 Co-emission of $NH_3$ and $NO_2$**

The peak in $NH_3$ concentrations in April occurs at the same time as a peak in $NO_2$ concentrations (though unlike $NH_3$, the April peak does not represent the highest annual atmospheric

concentration of $NO_2$).  For the month of April, total column $NH_3$ concentrations and tropospheric $NO_2$ concentrations integrated across the entire focal region are strongly correlated (r=0.78, p<0.0001). This simultaneous increase in atmospheric $NO_2$ and $NH_3$ concentrations provides additional, indirect support for a soil source of $NH_3$ (Figure 2, 6).  In addition to abundant field- and laboratory evidence that previously dry soils emit large pulses of NO following wetting (Davidson, 1992; Davidson $et$ $al$., 1991; Dick $et$ $al$., 2001; Meixner $et$ $al$., 1997), Jaeglé $et$ $al$. (2004) showed that this soil NO pulse is responsible for the highest concentrations of atmospheric $NO_2$ over the same region we focused on.  As with the seasonal pattern in $NH_3$ concentrations, seasonal variation in OMI-$NO_2$ tropospheric concentrations does not match that of the GFED4s $NO_2$ emissions from fires (Figure 6).  We believe that the simplest explanation for our box model results showing the strong correlation between tropospheric $NO_2$ and total column $NH_3$ concentrations in April is that the two pulses are the result of soil emissions triggered by the same environmental change—in this case, an increase in soil moisture following an extended dry period.

Like $NH_3$, concentrations of $NO_2$ also exhibit a peak in mid-March, but the peak lags a week behind the $NH_3$ peak (Figure 2).  Although NO emissions from soils can peak within hours after wetting (Davidson, 1992), in a laboratory setting using soils from the Mojave Desert, an NO pulse lagged 1 to 2 days behind an $NH_3$ pulse following experimental wetting (McCalley and Sparks, 2008).  The authors argue that this lag may be related to competition for $NH_4^+$ between $NH_3$ volatilization and nitrification, with volatilization generally outcompeting nitrification during the initial period following wetting.  An alternative or additional contributing factor to the lag may be related to the population dynamics of nitrifying bacteria.  Although the transcriptional response of nitrifiers to wetting can be very rapid (Placella and Firestone, 2013), at a population level nitrifiers are generally slow-growing (Robertson and Groffman, 2007), suggesting that populations in resource-limited environments may not be able to immediately take advantage of sudden large increases in resource availability.  For example, NO emission responses to inorganic N inputs—which arguably provides a similar immediate release from a resource limitation as wetting of dry soils for nitrifiers, particularly in relation to the presumed increase in $NH_4^+$ concentrations caused by wetting—exhibited pulse responses that lagged roughly a week behind fertilizer additions in Kenya (Hickman $et$ $al$., 2017).  In the case of the Sahel, nitrifier populations may not be able

to recover quickly from the extended dry conditions in March, and a lagged emission response of NO could be explained by the slow population-level response to the flush of mineralized N that follows wetting.

The March $NO_2$ peak is also smaller than the April $NO_2$ peak, and smaller than the April $NH_3$
peak. Field studies have observed larger pulses following the second rainfall of the season (Meixner *et al.*, 1997). The initial increase in soil moisture during March was modest (Figure 2), so it seems possible that soils dried out quickly, before populations of nitrifying bacteria grew large enough to trigger an NO pulse of the magnitude that occurs in April. Increased competition for available $NH_4^+$ by nitrifying bacteria in April may have limited the $NH_3$ pulse.

There is also a clear mis-match in the timing of the peak $NH_3$ concentrations, which occurred in March and April, and peak $NO_2$ concentrations, which occurred in May and June (Figure 2). We believe that several factors are likely contributing to the different temporal dynamics of $NH_3$ and $NO_2$ concentrations. First, to be clear, we argue that the rainfall events in March and April are indeed triggering pulses of both gases, illustrated in part by the strong correlation between the two gases during April
($r=0.78$, $p<0.0001$; Figure 2; for a strict comparison in which concentrations of $NO_2$ and $NH_3$ are calculated using only grid cells that have observations for both gases, see Figure S1, though the results are qualitatively similar to Figure 2). Temporal patterns of surface $NO_2$ concentrations observed by the INDAAF network may also support the presence of these smaller, early pulses (Figure S2). Although not explicitly investigated, we believe earlier studies of $NO_2$ pulsing in the Sahel that focus on emissions
during May and June include observations of similar early season pulses (Jaeglé *et al*. 2004, Hudman *et al*. 2012).

The smaller early season $NO_2$ pulses are potentially the result of the nitrifier population dynamics alluded to above. Specifically, because population growth is slow, nitrifying populations are smaller during March and April than they will become later in the growing season when they are released from
sustained environmental limitations on growth, and so nitrification rates remain lower in these early responses. In an earlier field study conducted in Davis, California, which also experiences distinct dry and rainy seasons, nitrifier population growth rates were very slow to respond to ammonium additions, roughly matching the temporal patterns of $NO_2$ emissions we observed in the Sahel (Okano *et al*. 2004).

In that study, population size did not increase during the first week following ammonium sulfate applications, and then rose by roughly 50% in the second week. Between the 13th and 39th day, however, the population roughly tripled. In the Sahel, nitrifying populations are presumably released from widespread water limitation around the beginning of May (Figure 2), and $NO_2$ concentrations grow to their maximum a little more than a month later. An additional point is that although the pulsing behaviour of $NH_3$ appears to diminish in May and June, total $NH_3$ emissions during those months are almost as elevated relative to the dry period as emissions during March and April (167 mg N $m^{-2}$ and 173 mg N $m^{-2}$ for each 61-day period, respectively; Figure 6). Consequently, it could be argued that May and June remain important months for both $NO_2$ and $NH_3$ emissions. For a presentation of modelled emissions calculated based only grid cells that have satellite observations for both gases, see Figure S3. This additional screening reduces annual modelled $NH_3$ emissions by roughly 5%, and reduces modelled emissions during the pulses by roughly 2-3% relative to modelled $NH_3$ emissions that do not take the presence or absence of $NO_2$ observations into account.

A second point is that both $NH_3$ volatilization and nitrification are dependent on $NH_4^+$ availability, and thus competition between these processes for $NH_4^+$ is likely to contribute to the observed temporal dynamics of atmospheric $NH_3$ and $NO_2$ concentrations. The increased competition for available $NH_4^+$ from growing nitrifier populations could contribute to the observed decline in $NH_3$ emissions between April and May. Earlier researchers have hypothesized that competition between $NH_3$ volatilization and nitrification for $NH_4^+$ is so extreme that the two processes cannot occur at the same time (Praveen-Kumar and Aggarwal, 1998). However, soil emissions of NO and $NH_3$ have been shown to be broadly coupled in the Sahel (Delon *et al.* 2018), and the few field and lab studies measuring soil emissions of both gases following wetting observe positive fluxes of each (McCalley and Sparks, 2008; Schlesinger and Peterjohn, 1991; Soper *et al.*, 2016), with $NH_3$ dominating emissions from desert soils (McCalley and Sparks, 2008; Schlesinger and Peterjohn, 1991) and NO dominating from grassland soils (Soper *et al.*, 2016). Our findings contrast with those of Soper *et al.*, as $NH_3$ appears to be the dominant species emitted during pulse events in March and April (Figure 6), though NO may dominate later in the season, during the period focused on by Jaeglé et al (2004). We expect that even were the two processes mutually

exclusive at the scale of a soil core or chamber, heterogeneity in soil properties at the pixel or regional scale would explain our observations of coinciding peaks of $NO_2$ and $NH_3$.

### 3.3 Magnitude and importance of soil $NH_3$ emission pulses

During the early growing season, atmospheric concentrations of $NH_3$ are roughly an order of magnitude higher than $NO_2$ (Figure 2). It is important to note that $NH_3$ retrievals generally have a higher error, and that our screening process may introduce a potential bias in that we permit retrievals with higher uncertainty if they are low concentrations; we also retain observations of negative concentrations. In addition, as mentioned earlier, compared to FTIR observations the IASI total columns are biased low by ~30% which varies per region depending on the local concentrations. From this perspective, our concentration and emission estimates can be considered conservative.

Because the magnitude of mean surface fluxes depends on the effective lifetime used in the simple box model, comparisons between the modelled fluxes of $NO_2$ and $NH_3$ are not straightforward. It is also important to note the simplifying assumption of a uniform atmospheric profile in the box model, which ignores any variation in vertical distribution. Still, we believe a uniform profile is a reasonable assumption, especially for soil fluxes. Each of these gases has relatively short lifetimes, and unlike fire plumes, are unlikely to be rapidly lofted to high altitudes following emission from soil. In addition there is very little variation in $NH_3$ distribution throughout the boundary layer in the assumed IASI profile. In general, our modelled emissions suggest that $NH_3$ is probably emitted at substantially higher rates from soils than NO during pulse events (Figure 6). This result is unexpected, given earlier observations that $NH_3$ tends to be the dominant source in highly alkaline desert soils (pH 9-11; McCalley and Sparks, 2008), whereas emissions from grassland soils with a more neutral pH, as might be expected in the Sahel, were dominated by NO, by roughly a factor of 10:1 (Soper *et al.*, 2016). This pulse produces $NH_3$ concentrations comparable in magnitude to the peak concentrations over many of the Earth's major biomass burning regions (Whitburn *et al.*, 2015). Indeed, $NH_3$ emissions during the date range February

29-March 16 and April 12-May 1, which cover the two emission peaks during that period, represented about one fifth of annual $NH_3$ emissions from our focal region in the Sahel (annual emissions of ~0.4 to 2 Tg N, and total pulse emissions of ~0.1 to 0.5 Tg N, depending on the effective lifetime assumed, but assuming that the mean effective lifetime during the pulses is equal to the mean effective lifetime throughout the year). It is important to note that in addition to uncertainty associated with the $NH_3$ retrievals, additional sources of bias and uncertainty—such as the use of an effective lifetime rather than explicitly accounting for deposition fluxes and chemistry, uncertainty in the value and variability in that lifetime, and biases in both the IASI and OMI retrievals—limit our ability to quantitatively constrain the surface $NH_3$ or $NO_2$ fluxes, or to make strict quantitative comparisons between them.

Within the Sahel, the March and April pulses are responsible for the highest concentrations of atmospheric $NH_3$ during 2008. The magnitude of $NH_3$ emissions during theses pulses are of a similar magnitude as emissions from biomass burning in north equatorial Africa (Whitburn *et al.* 2015). This substantial pulse of $NH_3$, and the co-occurrence of a pulse of $NO_x$, could be an important source of $PM_{2.5}$ in the region during the first half of the year. Secondary inorganic aerosols such as ammonium sulfate are formed in reactions involving $NH_3$, and ammonium nitrate aerosol formation requires both gaseous $NH_3$ and $NO_x$. In the Sahel, combustion sources of $NO_2$ are relatively small in March and April (Fig. 6), making soil NO emissions potentially more important in the formation of $PM_{2.5}$.

## 3.4 Comparison to surface observations

The INDAAF network (International Network to Study Deposition and Atmospheric Chemistry in Africa, http://www.indaaf.obs-mip.fr) provides monthly surface $NH_3$ deposition rates between 1998 to 2016 for seven sites in north equatorial Africa, both in and outside the Sahel. The work of Adon *et al.* (2010) presents an analysis of annual and seasonal variability of surface gases concentrations including $NO_2$ and $NH_3$ from the long-term monitoring INDAAF stations over the period 2000-2007. The sites in the Sahel (Agoufou, Mali, Banizoumbou, Niger, and Katibougou, Mali; figures 3b, 3c, and 3f, respectively) exhibit broadly similar seasonal patterns, with $NH_3$ concentration increases starting in April (or March in the 2008 case of Katibougou, Mali; Figure 3f), but this seasonal pattern is generally

absent in sites outside the Sahel (Figure 3d, 3e, 3g, and 3h). These seasonal patterns are broadly consistent with the pulsing dynamics observed by IASI in the Sahel, and the absence of pulsing outside the Sahel (Figure 3), though the peak concentrations observed at the surface tend to occur later than those observed by satellite in Banizoumbou and Katibougou.

5          The difference in the timing of peak $NH_3$ concentrations between the IASI observations and surface observations at Banizoumbou and Katibougou may be because of random variation and the effects of local influences, which are likely also responsible for variation among the three sites. In addition, deterministic variation in $NH_3$ emissions in the Sahel is likely important. These sites are located at latitudes near the northern boundary of our focal area, and could be influenced by emissions 10   from the north. In addition, we observe a north-south gradient in soil moisture, precipitation, and $NH_3$ concentrations during the pulse event in April (Figure 4), suggesting that pulsing is larger in the southern part of our focal area during this period. The onset of the rainy season tends to occur later in more northern latitudes, which would be consistent with higher $NH_3$ concentrations observed later in the year in Banizoumbou and Katibougou. Indeed, IASI observations of 1° grid cells over each of the 15   three INDAAF sites show better agreement with the surface observations, with peak NH3 concentrations occurring in May for all sites during 2008 (Figure S4).

        Other factors may contribute to variation in $NH_3$ concentrations observed across all the INDAAF sites. An earlier evaluation of seasonal patterns in the surface observation network found that emission patterns differ between wet savannah and dry savannah: dry season emissions tend to be higher in wet 20   savannah, where biomass burning dominates annual emissions (Adon *et al*., 2010). In dry savannah ecosystems, emissions are higher during the wet season, and are likely enhanced by the volatilization of N inputs from agro-pastoralism in the region, leading to high total N deposition fluxes (Adon *et al*., 2010). It is conceivable that regional differences in soil pH (Vågen *et al*., 2016) could also result in different rates of soil $NH_3$ emissions. In addition, higher leaf area index in wet savannahs and forest could result 25   in more interception of $NH_3$, reducing soil contributions to atmospheric concentrations during the wet season.

## 3.5 Later growing season NH$_3$ emissions

After the early season pulses, NH$_3$ concentrations and fluxes remain fairly elevated (Figure 2, 6); during this period of May through July, NO$_2$ concentrations and fluxes increase, becoming closer in magnitude to NH$_3$ (Figure 2, 6).  This pattern is consistent with earlier studies describing elevated concentrations of these gases in or near our focal study region for several months following the end of the biomass burning season (Jaeglé *et al.*, 2004; Whitburn *et al.*, 2015).  Jaeglé *et al.* (2004) attribute these elevated concentrations to the wetting of dry soils, and Whitburn *et al.* (2015) to increasing surface temperatures and the possible acidification of soils.  The mean start of the cropping season ranges from May to August in the focal region, based on NDVI-based observations of plant phenology (Vrieling *et al.*, 2011), suggesting that the bulk of elevated NO$_2$ and NH$_3$ in our modelled emissions occur after planting, and well into the rainy season.  Elevated emissions could be associated with soil disturbance from tillage (e.g., Yang *et al.*, 2015) or from the use of fertilizer inputs, which have been argued to be higher than generally acknowledged in the scientific and development literature (Cobo *et al.*, 2010; Sheahan and Barrett, 2017), though N additions in Sahelian countries such as Niger tend to be very low, even for farmers who use fertilizer (Masso *et al.*, 2017; Sheahan and Barrett, 2017).

## 3.6 Possible sources of error and bias

It is worth noting that the morning overpass time used in this analysis is likely to cause a further underestimation of daily NH$_3$ concentrations, as emissions would be expected to follow diurnal temperature variation and be higher in the afternoon (Van Damme *et al.*, 2015).  A more sophisticated inverse modelling approach (e.g., using a Bayesian approach in combination with an atmospheric chemical transport model) could provide firmer insight into the magnitude of emissions, as well as provide some insight into the magnitude of specific NH$_3$ sources.  Our analysis is subject to several additional sources of uncertainty: increased cloud cover during the rainy season tends to result in fewer observations than during the dry season, so our regional means are based on different numbers of observations at different times of year.

Although the explanatory power of the linear relationship between soil moisture and emissions in the early part of the rainy season is relatively low, this would be expected in part due to the complex processes which vary over time. Multiple studies have shown that rewetting of dry soils results in lower emission pulses for the same level of water addition (Davidson, 1992; Davidson *et al.*, 1991), including of $NH_3$ emissions (Soper *et al.*, 2016). It also seems plausible that there are threshold effects in which an initial increase in soil moisture may simply need to be large enough to activate dormant microbial communities and/or cause a flush of labile microbial N to trigger an emissions pulse, such that the pulse response might best be described with a piecewise function. Asynchrony between plant and microbial activity during soil wet up and dry down (Collins *et al.*, 2008), and different activation thresholds for microbial and plant responses to precipitation (Dijkstra *et al.*, 2012) may also play a role in determining the amount of available $NH_4^+$ for volatilization at different times in the early rainy season.

### 3.7 Conclusion

Satellite measurements of trace gases—although not without their limitations—provide a powerful tool for understanding global and regional atmospheric composition, and for gaining insights into the controls over nitrogen cycling and trace gas emissions, particularly for regions where other types of measurements are scarce. With daily and global coverage and additional satellite observed variables such as precipitation, soil moisture, biomass burning emissions, and tropospheric $NO_2$ concentrations, it is possible to evaluate specific mechanisms behind the seasonality of trace gas emissions. In an evaluation of the Sahel during 2008, we find that $NH_3$ concentrations are elevated during March and April, a period when biomass burning emissions are absent, but when tropospheric $NO_2$ concentrations exhibit similar temporal dynamics. We further find that the increase in $NH_3$ concentrations is positively correlated to changes in soil moisture at the start of the rainy season. Using a simple box model, we estimate that average emissions for the entire Sahel are between 2 and 6 mg $NH_3$ $m^{-2}$ $day^{-1}$ during peaks of the observed pulses, though note that these estimates are subject to substantial bias and uncertainty. We conclude that the Birch effect is an important and geographically broad driver of $NH_3$ emissions, and an important component of the N cycle in the Sahel.

**Acknowledgements:** The IASI-NN observations were created by the atmospheric spectroscopy group at ULB (Spectroscopie de 645 l'Atmosphère, Service de Chimie Quantique et Photophysique, Université Libre de Bruxelles, Brussels, Belgium), and were obtained from http://espri.aeris-data.fr/etherTypo/index.php?id=1727&L=1. We would like to thank Simon Whitburn, Martin Van Damme, Lieven Clarisse and Pierre Francois Coheur for the retrieval product. JEH would also like to thank Shelly van der Graaf for her assistance in interpreting the IASI data.

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

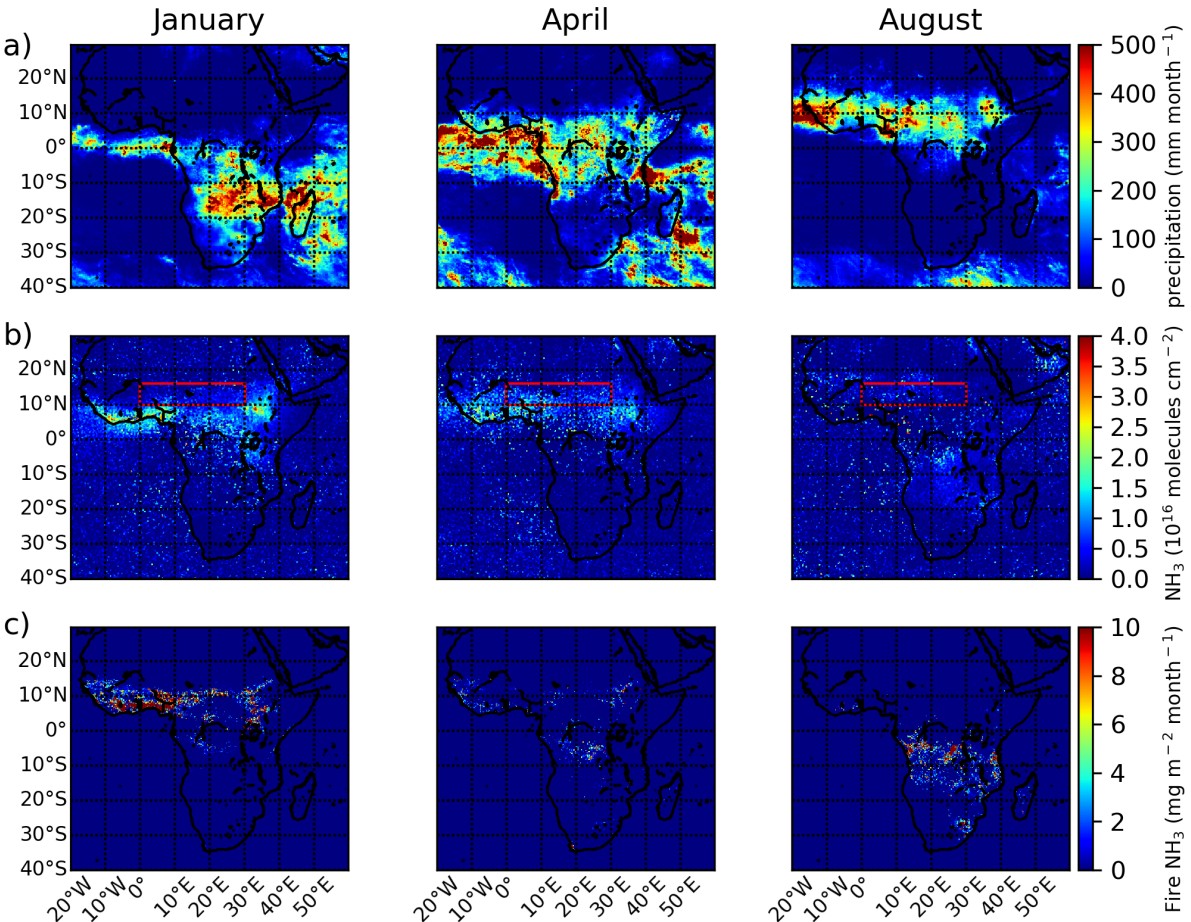

Figure 1. *Seasonality of precipitation and NH₃ emissions in Africa during 2008.* Monthly TRMM precipitation (a), IASI-NH$_3$ concentrations (b), and GFED4s NH$_3$ emissions (c) over Africa in January, April, and August, 2008. TRMM precipitation is presented in mm month$^{-1}$, IASI-NH$_3$ concentrations in 10$^{16}$ molecules cm$^{-2}$, and GFED4s NH$_3$ emissions in mg m$^{-2}$ month$^{-1}$.

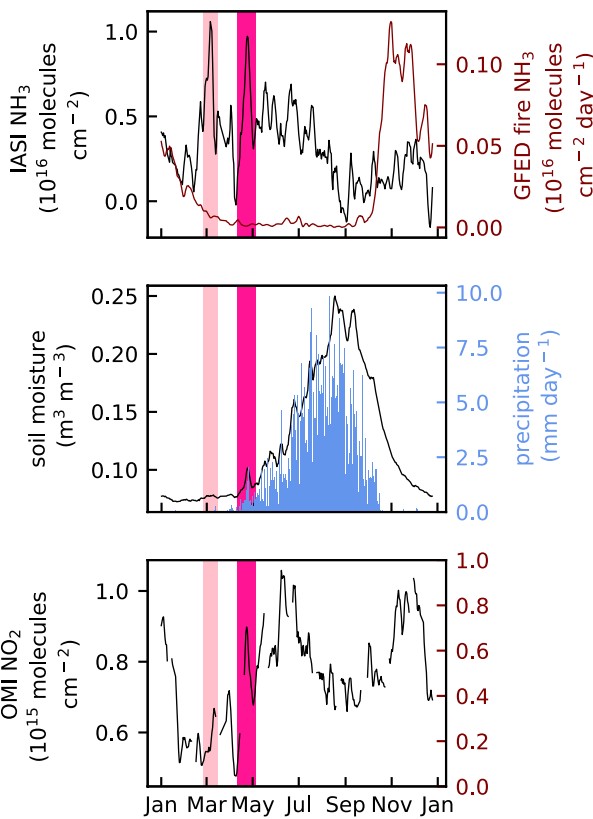

Figure 2: *Early growing season NH₃ pulses temporally associated with changes in soil moisture and with NO₂ pulses over the focal study region in the Sahel during 2008*. Top panel: Daily atmospheric NH₃ concentrations from IASI and NH₃ biomass burning emissions from GFED4s. Middle panel: ESA-CCI soil moisture and TRMM precipitation. Bottom panel: atmospheric NO₂ concentrations from OMI. Putative soil emission pulses in March and April are highlighted in bright and dark pink, respectively. Mean NO₂ concentrations were calculated using values of 0.25° grid cells within the study region for which NH₃ observations were also present. GFED4s emissions were converted from a mass-based to molecule-based flux to allow comparison with the IASI retrievals; note the different scales for the left and right y axes of the top panel.

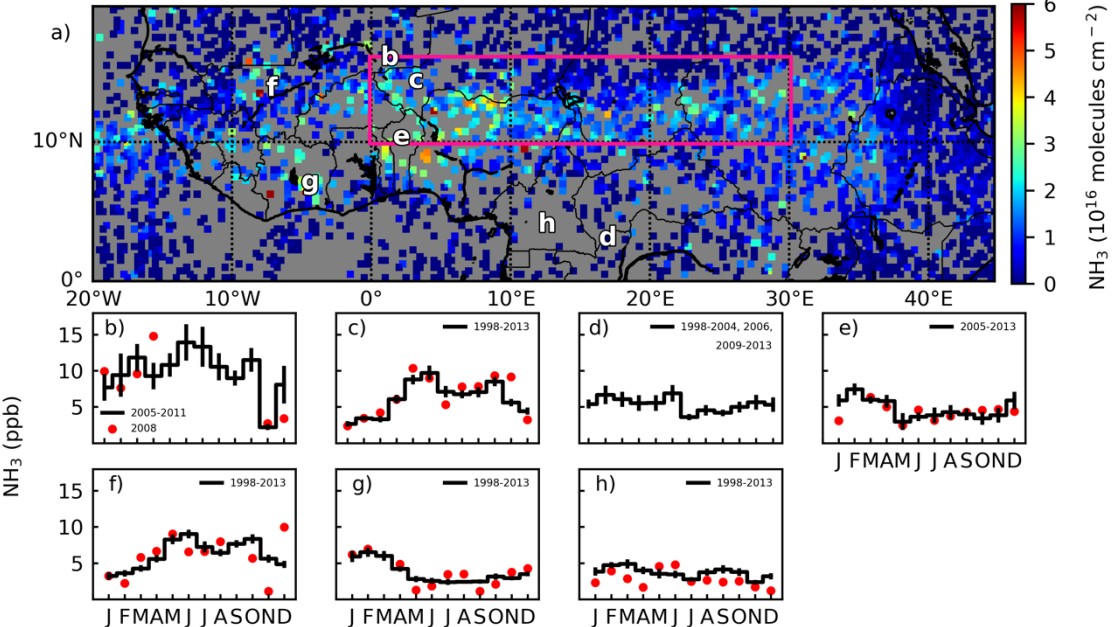

Figure 3. *NH₃ pulsing over the Sahel in April, and geographic variation in the seasonality of surface NH₃ concentrations over north equatorial Africa.* (a) Observations by IASI during April 25 to April 28, 2008 reveal elevated mean atmospheric NH₃ concentrations specifically over the Sahel region (b-h). Monthly NH₃ gas concentrations from sites in the INDAAF network; black lines represent the multi-year mean and standard error for each site, and red dots represent the 2008 value. Data are presented for Agoufou, Mali (b), Banizoumbou, Niger (c), Bomassa, Congo (d), Djougou, Benin (e), Katibougou, Mali (f), Lamto, Côte d'Ivoire (g), and Zoetele, Cameroon (h).

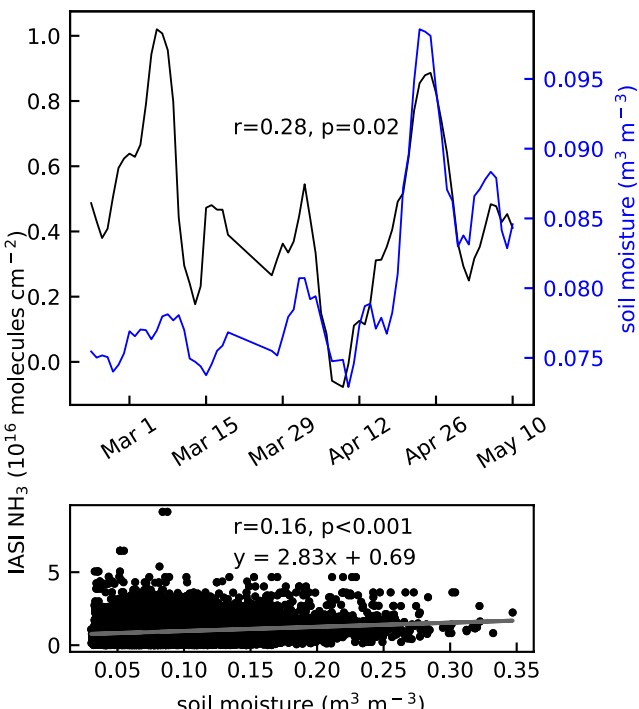

Figure 4. *Correlations between soil moisture and atmospheric NH3 concentrations observed by IASI over the focal study region in the Sahel during the start of the rainy season in 2008.* Top panel: 5-day running mean of daily $NH_3$ concentrations and daily soil moisture for the focal study region from mid-February through the end of April, 2008. Bottom panel: scatterplot of soil moisture versus atmospheric $NH_3$ concentration for each 0.25° grid cell in the study region during April, 2008. In both panels, soil moisture data are included only for those 0.25° grid cells where $NH_3$ observations are available for the same day.

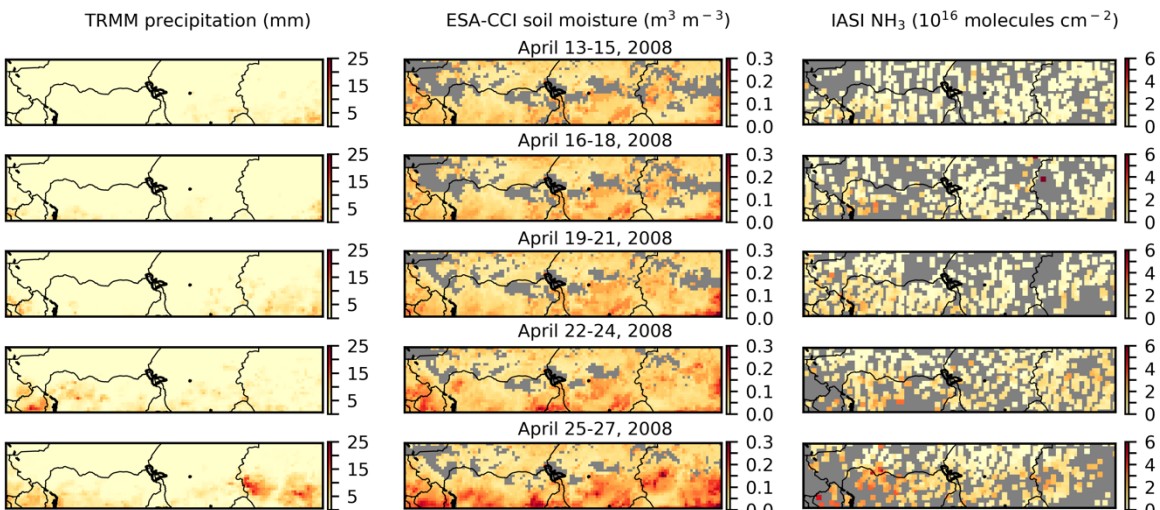

Figure 5. Maps of changing precipitation, soil moisture, and atmospheric NH₃ concentrations for the focal region of the Sahel during the second half of April, 2008. 3-day averages from April 13 through April 27 are presented for precipitation from TRMM (left column), soil moisture from ESA-CCI (middle column) and atmospheric NH₃ concentrations for acceptable retrievals from IASI (right column).

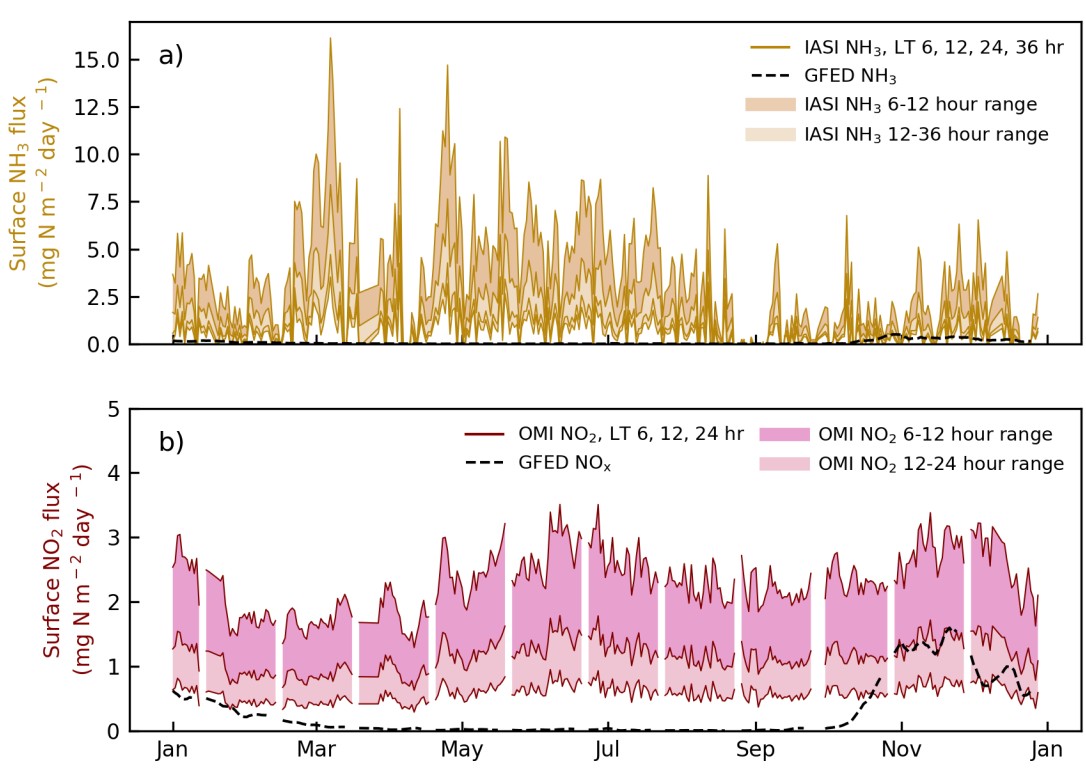

Figure 6. *Comparison of daily total surface fluxes and biomass burning emissions of a) NH₃-N and b)*
*NO₂-N for the focal study region in the Sahel during 2008.* Total surface fluxes are estimated from IASI
NH₃ and OMI NO₂ observations using a simple box model and assuming effective lifetimes of 12, 24, or
36 hours for NH₃ and of 6, 12, or 24 hours for NO₂. Fire emissions are taken from the GFED4s database.
Modelled NO₂ emissions were calculated using values of 0.25° grid cells within the study region for
which NH₃ observations were also present. GFED4s emission means were calculated using 0.25° grid
cells that matched those used in the modelled emissions for the respective gas. Note the difference in
scales, and that shorter effective lifetimes result in higher modelled emissions.

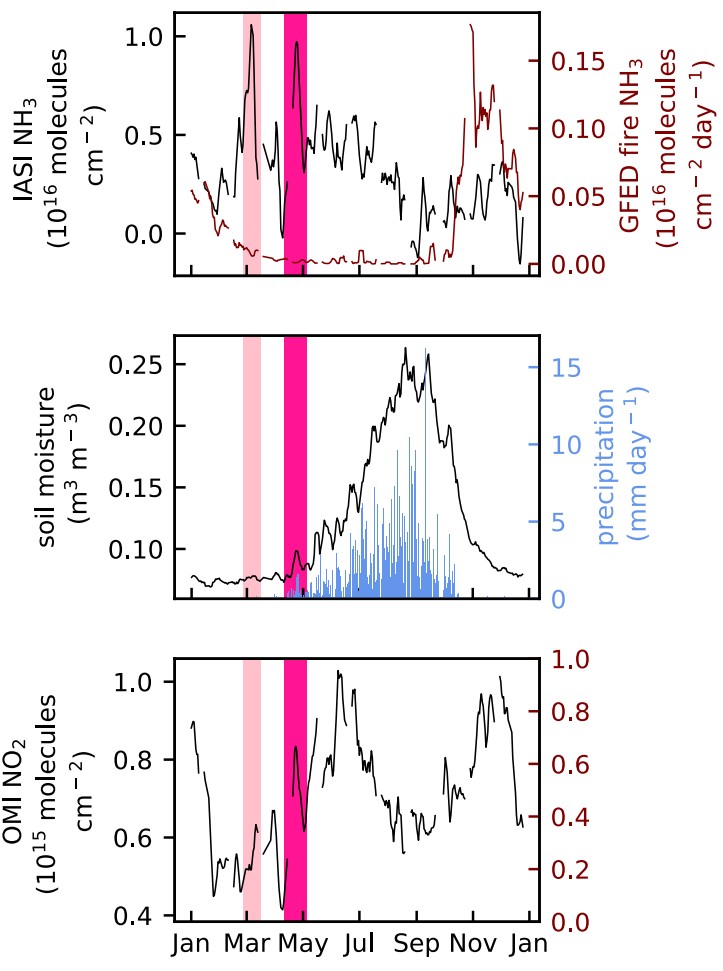

Figure S1: *Early growing season NH₃ pulses temporally associated with changes in soil moisture and with NO₂ pulses over the focal study region in the Sahel during 2008.* Top panel: Daily atmospheric NH₃ concentrations from IASI and NH₃ biomass burning emissions from GFED4s. Middle panel: ESA-CCI soil moisture and TRMM precipitation. Bottom panel: atmospheric NO₂ concentrations from OMI. Putative soil emission pulses in March and April are highlighted in bright and dark pink, respectively. The means of all variables presented were calculated using values of 0.25° grid cells within the study region for which both NO₂ and NH₃ observations were present. GFED4s emissions were converted from a mass-based to molecule-based flux to allow comparison with the IASI retrievals; note the different scales for the left and right y axes of the top panel.

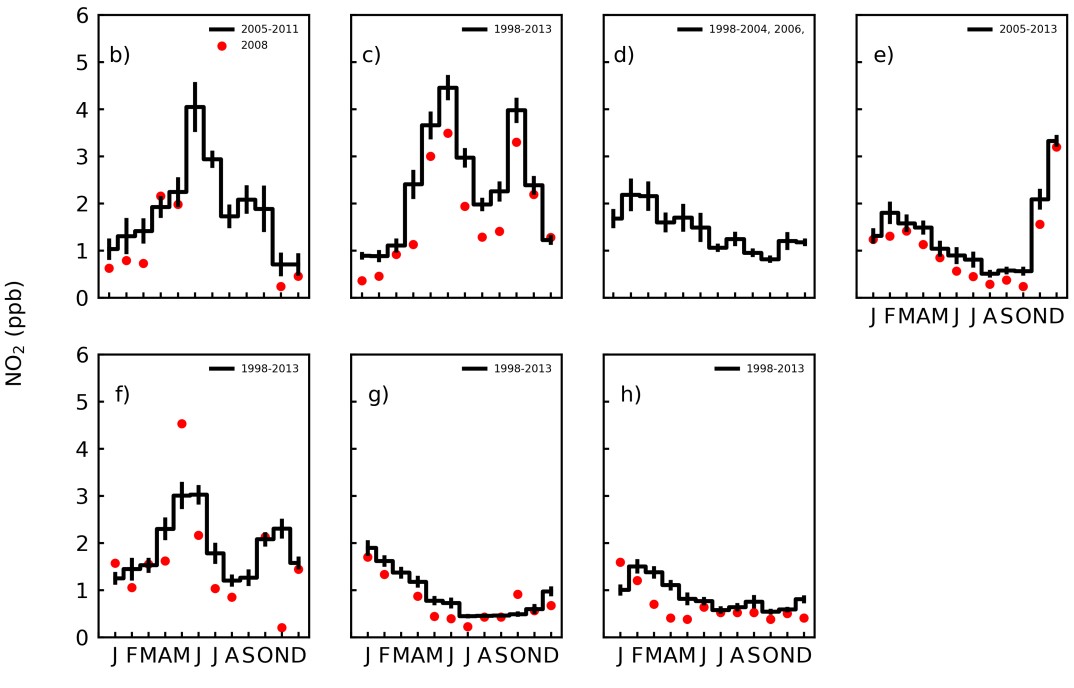

Figure S2. *Geographic variation in the seasonality of surface NO₂ concentrations over north equatorial Africa*. Monthly NO₂ gas concentrations from sites in the INDAAF network; black lines represent the multi-year mean and standard error for each site, and red dots represent the 2008 value. Data are presented for Agoufou, Mali (b), Banizoumbou, Niger (c), Bomassa, Congo (d), Djougou, Benin (e), Katibougou, Mali  (f), Lamto, Côte d'Ivoire (g), and Zoetele, Cameroon (h).  Note that there is no panel (a) in this figure, so that the site labels match those in Figure 3.

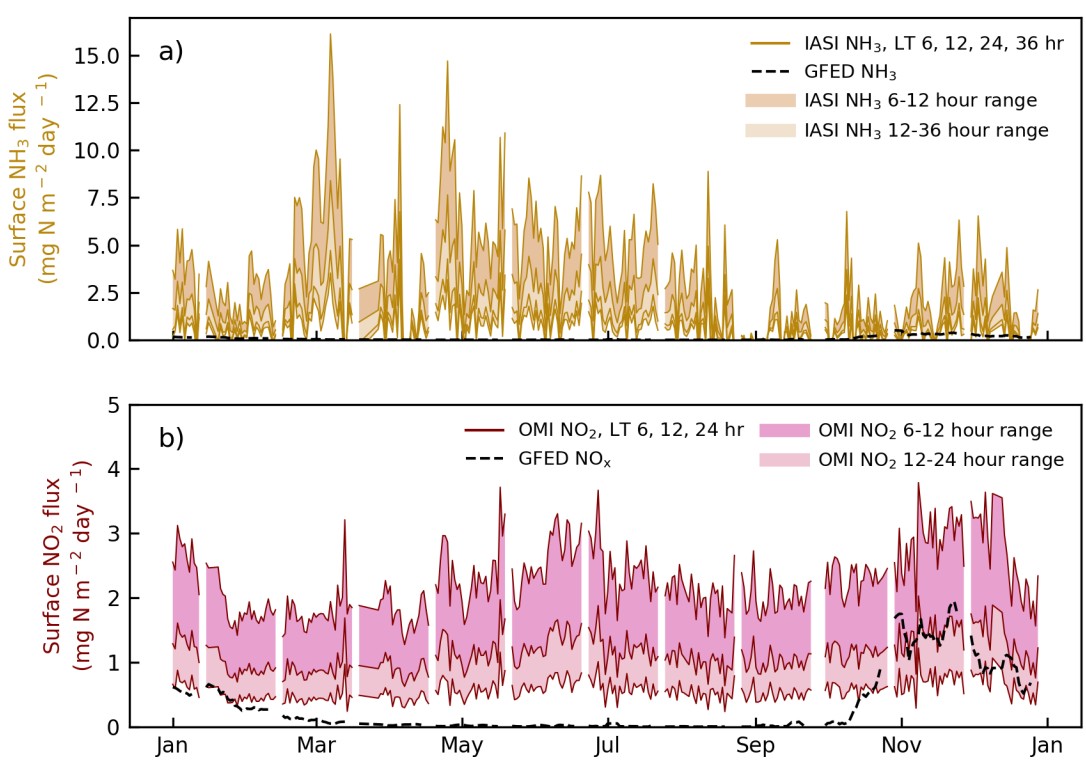

Figure S3. *Comparison of daily total surface fluxes and biomass burning emissions of a) NH₃-N and b) NO₂-N for the focal study region in the Sahel during 2008*. Total surface fluxes are estimated from IASI NH$_3$ and OMI NO$_2$ observations using a simple box model and assuming effective lifetimes of 12, 24, or 36 hours for NH$_3$ and of 6, 12, or 24 hours for NO$_2$. Fire emissions are taken from the GFED4s database. Modelled and GFED4s mean emissions were calculated using values only of 0.25° grid cells within the study region for which both NO$_2$ and NH$_3$ observations were present. Note the difference in scales, and that shorter effective lifetimes result in higher modelled emissions.

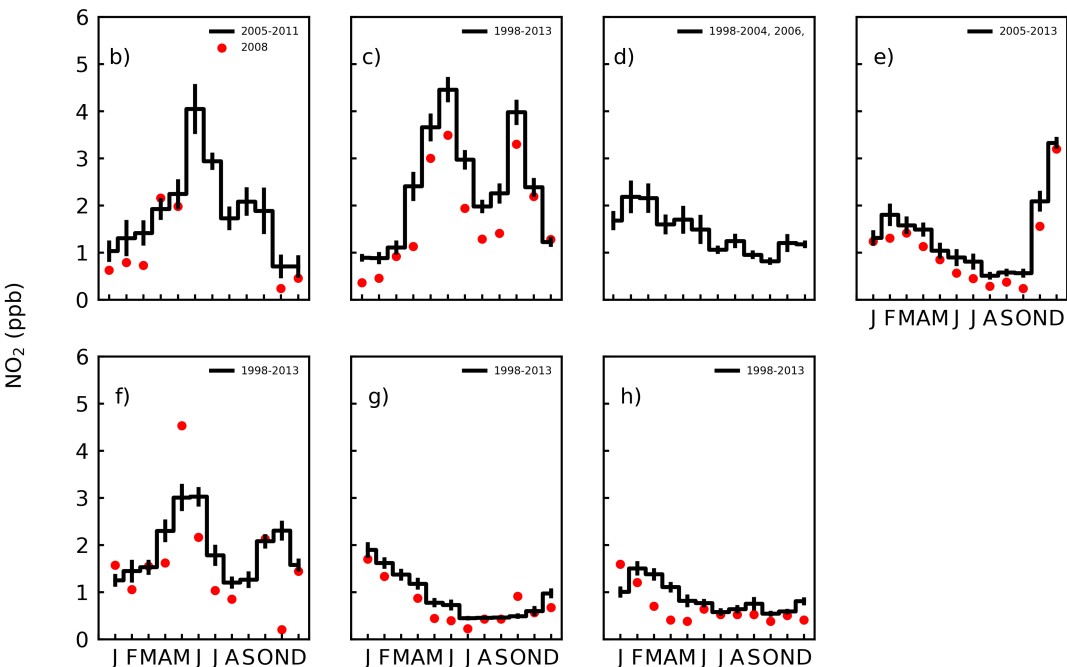

Figure S4. *Monthly means of IASI observations of atmospheric NH₃ concentrations for 1° grid cells centered over Agoufou, Mali, Banizoumbou, Niger and Katibougou, Mali in 2008.* Surface observations for each site are presented in Figure 3b, 3c, and 3f, respectively.

