# Peer review of "Satellite evidence of substantial rain-induced soil emissions of ammonia across the Sahel"

_Atmospheric Chemistry and Physics, 2018_

## Author Comment (AC1) · 1 Aug 2018

The lead author inadvertently omitted the following acknowledgement from the submitted manuscript:

"The IASI-NN observations were created by the atmospheric spectroscopy group at ULB (Spectroscopie de 645 l'Atmosphère, Service de Chimie Quantique et Photophysique, Université Libre de Bruxelles, Brussels, Belgium), and were obtained from http://espri.aeris-data.fr/etherTypo/index.php?id=1727&L=1. We would like to thank Simon Whitburn, Martin Van Damme, Lieven Clarisse and Pierre Francois Coheur for the retrieval product."

---

## Referee Comment (RC1) · Anonymous Referee #1 · 2 Aug 2018

This manuscript examines pulses of NH3 observed by the IASI-A instrument on MetOp-A over the Sahel. These pulses occur at the start of the rainy season and appear to correspond to increased biochemical activity in soils induced by rainfall. The paper is well written, the analysis is clear, and the results are very interesting. My main concern is that the timing of the enhancements in OMI NO2 (May-June), IASI NH3 (March-April), and surface NH3 (May-June) appear to be different. I elaborate on this below.

The timing of the NH3 peaks seems to be different than those of NO2. Several studies (for example Jaeglé et al., 2004; Hudman et al., 2012) have examined satellite NO2

[Figure]

pulses over the Sahel, finding that the pulses take place sometime in May-June, corresponding to the arrival of the first rains after the dry season over various regions of the Sahel. Figure 2, shows the same pattern of the largest OMI NO2 enhancement taking place in June. For NH3, the authors focus mostly on March and April. Some discussion of why the timing would be different in terms of pulses of NO2 and NH3 would be useful to include in the manuscript (for example in section 3.2.3 discussing the co-emissions of NH3 and NO2). Does it have to do with different population dynamics of the various bacteria and their response to subsequent wetting events? Interestingly, Figure 3 shows that surface observations do indicate maximum NH3 mixing ratios in May-June (at least for sites b, c, f), consistent with the OMI NO2 pulses. Why would surface observations of NH3 show a different seasonality than the satellite observations of NH3? Is the increasing cloud cover affecting the number and quality of the retrievals in May-June? Is the later June NH3 pulse masked by clouds?

Minor comments. - Throughout the manuscript (including the abstract). The authors refer to NO2 as 'nitric dioxide'. It should be nitrogen dioxide.

- Equation 1. It isn't clear how this is used for low concentrations. Does the equation mean that even if the error is above 100%, if the concentrations are low enough then the retrievals are kept? Also, it would be useful to have units after 5x1015 (I assume it is molec/cm2)

- Equation 3. There is an extra space before the tx (effective lifetime), also, x should be subscripted.

---

## Referee Comment (RC2) · Anonymous Referee #2 · 6 Aug 2018

I have read the comments from the first reviewer and agree that this is a clear, well written and very interesting paper. I also agree that the offset between the NH3 and NO2 peaks should be discussed in slightly more depth. What does the INDAAF NO2 data show?

I do think some clarification from the authors would be helpful on the following issues:

Page 7, line 7: please add a sentence of two on the regridding technique used.

Page 7, line 11: what is the IASI detection limit in the Sahel?

Page 8, line 24: it appears that a uniform profile is assumed here. Is this really a good

assumption?

Page 11, line 11: I don't see the increase in fire emissions in late March

Page 12, line 25: It could be helpful to rewrite this as: " except during the biomass burning season"

Page 13, line 23: at what scale where these correlations calculated: 0.25 deg or over the entire box?

Page 15, line 22: possibly repeat that IASI NH3 appears to be biased low

Page 17, line 17: only NO2 concentrations become comparable to NH3, not the fluxes

Page 18, line 8: a more sophisticated inverse modeling of the emissions? This should be made clearer, and should suggest an approach

Page 18, line 10: the increased cloud cover during the rainy season could certainly mask increased NH3. Cloud cover would also impact OMI NO2, but possibly not to the same degree. If at all possible, the authors should discuss this.

Minor comments:

Page 4, line 27: the second sentence does not follow from the first

Page 7, line 7: IASI data are only used if the pixels are 75% cloud-free, while OMI data is included only if the cloud-cover is less than 30%. While these statements are almost equivalent, it would be clearer to say 75% cloud-free and 70% cloud-free, or cloud cover less than 25% and cloud cover less than 30%.

Page 9, line 25: "burned area and fuel consumption in savannas are"

---

## Author Response (AR1)

REVIEWER 1:

This manuscript examines pulses of NH3 observed by the IASI-A instrument on MetOp- A over the Sahel. These pulses occur at the start of the rainy season and appear to correspond to increased biochemical activity in soils induced by rainfall. The paper is well written, the analysis is clear, and the results are very interesting. My main concern is that the timing of the enhancements in OMI NO2 (May-June), IASI NH3 (March-April), and surface NH3 (May-June) appear to be different. I elaborate on this below.

The timing of the NH3 peaks seems to be different than those of NO2. Several studies (for example Jaeglé et al., 2004; Hudman et al., 2012) have examined satellite NO2 pulses over the Sahel, finding that the pulses take place sometime in May-June, corre- sponding to the arrival of the first rains after the dry season over various regions of the Sahel. Figure 2, shows the same pattern of the largest OMI NO2 enhancement taking place in June. For NH3, the authors focus mostly on March and April. Some discus- sion of why the timing would be different in terms of pulses of NO2 and NH3 would be useful to include in the manuscript (for example in section 3.2.3 discussing the co-emissions of NH3 and NO2). Does it have to do with different population dynamics of the various bacteria and their response to subsequent wetting events? Interestingly, Figure 3 shows that surface observations do indicate maximum NH3 mixing ratios in May-June (at least for sites b, c, f), consistent with the OMI NO2 pulses. Why would surface observations of NH3 show a different seasonality than the satellite observa- tions of NH3? Is the increasing cloud cover affecting the number and quality of the retrievals in May-June? Is the later June NH3 pulse masked by clouds?

This is an excellent question, and we are happy to add more discussion regarding what is happening. In the following, we've broken our response into sections for each major question, and then present the new discussion added to the revised manuscript.

1) Why do we see NH3 pulses in March and April when Jaegle et al 2004 and Hudman et al 2012 find that "pulses take place sometime in May-June, corresponding to the arrival of the first rains after the dry season over various regions of the Sahel?" Our first point is that we believe we have presented evidence that soil NO pulses are occurring in response to the same increases in soil moisture that cause the soil NH3 pulses in March and April 2008, and it seems to us that similar pulsing is observed in both the Jaegle et al 2004 and Hudman et al 2012 papers. As noted on page 14 line 11, satellite observations of NO2 and NH3 concentrations are strongly correlated for the month of April 2008 (r=0.78, p<0.0001), suggesting that the increases in soil moisture are leading to pulsed emissions of both gases. The following figure may help to illustrate the correlation; it reproduces the data included in the top and bottom panels of figure 2 in a single panel (note that each date's means are calculated using only grid cells for which both NO2 and NH3 observations were present):

[Figure]

The correspondence of the NO2 and NH3 peaks in April is fairly clear here. We also interpret the March NH3 peak to be accompanied by an NO2 peak that emerges about a week later. We discuss the apparent lag between emission pulses of NH3 and NO2 in March on page 14, line 24 and following, and expand upon it in the revised manuscript (details below). The general idea is that populations of nitrifying bacteria, which are generally responsible for the vast majority of NO emissions from soils, are likely dormant when this March increase in soil moisture occurs. Nitrifiers are notoriously slow growing, resulting in the lag between the pulse of NH3 and NO (in California field soils, no increase in nitrifying populations was observed for the first seven days following fertilizer applications (Okano et al 2004 Appl Env Microbiol.); nutrient additions are physically quite different from wetting, but both do represent release from an environmental limitation on growth). Lags are often observed between wetting or fertilization of soils and subsequent NO emissions from soils in experimental studies. This increase in soil moisture in March is also very modest, which may have limited the magnitude of the NO response more than that of the NH3 response—that is, the increase in soil moisture may have been large enough to trigger a flush of N mineralization and more-or-less immediate NH3 volatilization, but the soils may have dried again before the nitrifying populations were able to really kick into gear (or grow in size), limiting the magnitude of the NO pulse.

Something else to note is that Hudman et al.'s (2012) analysis of atmospheric NO2 concentrations in 2006 uses a 2mm rainfall threshold to define the start of the rainy season. That threshold that is indeed met for much of the Sahel in April of that year, as illustrated in their Figure 2, and thus the pulsing that Hudman et al observe includes pulsing in April. Those patterns are arguably broadly consistent with what we observe here, and the increase in soil moisture we observe in March occurs with average rainfall well under 2 mm (figure 2). The NO2 enhancements that Hudman et al observe are increases roughly between 1 and 1.5 times higher than concentrations 5 days previous to the first rainfall event for a given pixel, which are similar to the enhancements in NO2 we observe during both the March and the April pulse events. (We recognize that Hudman et al do not identify a lagged response, though again, that lag occurred in March, in response to average rainfall below 2 mm.) Jaegle et al focus on enhancements in June, but if you look at the annual data presented in their figure 2, the overall pattern is quite similar to what we observe in 2008: two early modest rainfall events in April and May appear to be accompanied by pulses in NO2 concentration.

2) If it is agreed that there is, in fact, pulsing of both NO2 and NH3 in response to the same increases in soil moisture in March and April, then the question becomes "why do NO2 concentrations reach their maxima in May and June, whereas the NH3 concentrations reach maxima in March and April?" This again is a good question. We believe it is probably related to the different mechanisms responsible for emissions of NH3 and NO from soils, and potentially in part to the dependence of both processes on ammonium ($NH_4^+$) availability. NO is primarily produced biologically during two processes: the transformation of NH4+ to nitrate during nitrification, and the transformation of nitrate to N2 during denitrification. NH3 is produced abiotically when ammonium is deprotonated. We mentioned above the slow-growth of nitrifying populations; generation times of 373 hours were observed for population responses of ammonia oxidizers to ammonium sulfate applications in a maize field in Davis, California (Okano et al 2004 Appl Env Microbiol.). In that example, population size did not increase at all during the first week following the ammonium sulfate applications, and rose by roughly 50% during the second week. Between the 13th and 39th day, however, the population roughly tripled. This is just one study, and it is looking at the effects of fertilizer application rather than of wetting (although it was conducted in the early growing season in a region with distinct rainy and dry seasons, and both wetting and fertilization represent a release from environmental constraints on population growth), but the slow rate of nitrifying population growth observed in Davis matches the patterns observed over the Sahel very well: nitrifying populations are presumably released from widespread water limitation starting around the end of April, and NO concentrations grow to their maximum a little more than a month later.

A second point here is that both NH3 volatilization and nitrification are dependent on the availability of NH4+, and thus we can potentially explain why NH3 concentrations decline somewhat when NO2 concentrations increase: growing nitrifying populations represent more competition for available NH4+. Some researchers have suggested that this competition is so steep that the two processes are mutually exclusive (Praveen-Kumar and Aggarwal, 1998). As environmental conditions become more

accommodating for nitrifying populations, NH3 emissions could be expected to necessarily decrease as NO emissions increase, all else being equal. However, it is interesting to note that at 167 mg N m$^{-2}$, the mean total NH3 emissions per unit area for the 61-day period in May and June are still quite elevated relative to the dry season, just a slight decline from the 61-day period in March and April (173 mg N m$^{-2}$). So it would appear that both periods are important for NH3 emissions, although in March and April, the pulsing dynamics appear to be much more distinct and important.

3. Could changing NH3 observation counts be responsible for the mismatch? It does not appear so. Although there is a drop in observation counts in July and August, for the period of March through the end of June (green points in the plot below), observation counts are relatively constant. In addition, in our original correlation analysis, we only used grid cells that included both NO2 and NH3 observations; in our presentations in figures 2 and 6, we included NO2 grid cells that also had NH3 observations, and so any change in NH3 observation counts would have affected the temporal patterns of both gases equally. In this revision, we include additional analyses that consider only grid cells where both NH3 and NO2 observations, with no qualitative effect on the results (see response to Reviewer 2's comment regarding Page 18, line 10 below).

[Figure]

4. Why is there an apparent mismatch between INDAAF surface observation sites and satellite observations of NH3? Another good question. Spatial variation in the timing and magnitude of NH3 emissions and concentrations is to be expected, and we believe that the basic issue is that these sites are not representative of the mean for our focal region of the Sahel. When examining patterns of monthly mean NH3 concentrations for 1° grid cells centered on each of the three INDAAF Sahel sites (b, c, and f), we see that the overall pattern for 2008 observed by IASI and at the surface is in better agreement with the surface observations than the regional mean of IASI observations is.

[Figure]

In these satellite observations of the atmosphere over each of the surface sites, the increase in spring NH3 concentrations occurs later in the year than the regional mean, with peak NH3 concentrations occur in May for each location. This is perhaps not surprising, since our focal region is characterized by a gradient of increasing mean annual precipitation from north to south, combined with generally earlier onset of the rainy season in the south than the north. Hence, we may expect to see earlier and/or larger NH3 emissions from the southern than the northern part of the region on average (e.g., Figure 5 in the original manuscript), and the INDAAF Sahel sites are largely in the northern part of the latitudinal range we examine.

[revised manuscript text omitted]

Minor comments. - Throughout the manuscript (including the abstract). The authors refer to NO2 as 'nitric dioxide'. It should be nitrogen dioxide.

Thank you for catching this error, though we want to point out that it occurred only once (in the abstract: page 1, line 20). Throughout the manuscript, we use nitric oxide (NO) when discussing soil emissions, and nitrogen dioxide (NO2) when discussing OMI observations. We recognize that this could lead to some confusion, but also believe it is a precise description of the processes and observations discussed in the manuscript (suggestions for improving clarity are welcome). We have made a few changes for improved accuracy and to try to minimize any confusion. We also include the following statement to highlight our use of NO, NO2, and NOx in the manuscript:

"Note that in this paper, NO is used in discussions of soil emissions specifically; since satellite observations are of $NO_2$, we use $NO_2$ when discussing those observations, and $NO_2$ or $NO_x$ when discussing modelled surface emissions based on those observations." (Page 5, line 17)

Some example changes; in these examples, "NO2" was formerly "NO" and "peak" was formerly "pulse" (new text in bold italics):

"The March *NO₂ peak* is also smaller than the April *NO₂ peak*, and smaller than the April NH₃ *peak*."

"We expect that even were the two processes mutually exclusive at the scale of a soil core or chamber, heterogeneity in soil properties at the pixel or regional scale can explain our observations of coinciding *peaks* of *NO₂* and NH₃."

"In general, however, our modelled emissions suggest that NH₃ is probably emitted at substantially higher rates than NO *or NO₂* during pulse events,"

"limit our ability to quantitatively constrain the surface NH₃ or **NO₂** fluxes, or to make strict quantitative comparisons between them."

- Equation 1. It isn't clear how this is used for low concentrations. Does the equation mean that even if the error is above 100%, if the concentrations are low enough then the retrievals are kept? Also, it would be useful to have units after 5x1015 (I assume it is molec/cm2)

It does indeed mean that if the concentrations are low enough then the retrievals are kept. The errors are based on a statistical analysis of variability in the retrieval parameters observed over a clean patch in the pacific (Whitburn 2016). But under the right conditions, it is possible that a retrieval is correct while the error (being slightly fixed) is too high, and so the equation allows for retaining these low concentration retrievals. This also resolves the issue of biasing high when removing all small values (as most have a relative error above 100). And thank you: we have added the units to the equation.

Whitburn, S., Van Damme, M. and Clarisse, L.: A flexible and robust neural network IASI-NH3 retrieval algorithm, J. Geophys. Res.-Atmos., 121, 6581–6599, doi:10.1002/(ISSN)2169-8996, 2016

- Equation 3. There is an extra space before the tx (effective lifetime), also, x should be subscripted.

We have made the changes

REVIEWER 2:

I have read the comments from the first reviewer and agree that this is a clear, well written and very interesting paper. I also agree that the offset between the NH3 and NO2 peaks should be discussed in slightly more depth. What does the INDAAF NO2 data show?

INDAAF NO2 observations (as illustrated in the following figure) follow a broadly similar pattern to the OMI NO2 observations, with the highest concentrations occurring in May and June in Banizoumbou (c), Katibougou (f), and possibly in Agoufou (b), though data for June 2008 are not available.  The pattern in Agoufou and Banizoumbou are interesting, with elevated concentrations in April (and March for Banizoumbou) in 2008, but with peak NO2 concentrations occurring in May and June (Banizoumbou) or June and July (Agoufou long-term average).  These data suggest the presence of smaller early season NO pulses in these sites.   Katibougou is also very interesting: the concentration peak occurs in May, with a substantial decline in June.  We include the following figure as SI for the revised manuscript.

[Figure]

Agoufou, Mali (b), Banizoumbou, Niger (c), Bomassa, Congo (d), Djougou, Benin (e), Katibougou, Mali  (f), Lamto, Côte d'Ivoire (g), and Zoetele, Cameroon (h).

It appears to us that the patterns here reflect both the possibility of early season pulses in Agoufou and Banizoumbou, and also the presence of substantial variability in the timing of peak concentrations among the three sites located within the Sahel. That spatio-temporal variability in the timing of peak NO2 concentrations would seem to highlight the potential importance of local influences which can contribute to different patterns between sites, and potentially between surface and satellite observations.

I do think some clarification from the authors would be helpful on the following issues:

Page 7, line 7: please add a sentence of two on the regridding technique used.
   We have added the additional clarifying sentence (Page 7, line 14): "*Specifically, we calculated the concentration for a given grid cell as the mean of all elliptical IASI footprints for which the corners of the grid cell were within the footprint.*"
Page 7, line 11: what is the IASI detection limit in the Sahel?

This is a difficult question to answer as there aren't any hourly measurements available in the Sahel region, which makes it hard to compare the conditions with the satellite observations. However, the Sahel is a region with high thermal contrast (in contrast to, say, northern Europe), which makes it a region with generally optimal conditions for IASI. From comparisons with measurements under conditions of high thermal contrast, IASI seems to be able to reliably observe down to 1 to 2 ppb at the surface, which translates roughly into ~2e15/5e15 depending on the boundary layer height and other factors. This level of sensitivity is also seen in Van Damme et al., 2015 and Dammers et al., 2016, which found acceptable correlations between IASI and independent observations, even for background-like concentrations. We have added the following text (page 7 line 9):

"Given the absence of hourly observations in the Sahel, the detection limit of IASI is difficult to determine with certainty. However, the region experiences high thermal contrast, and IASI seems to be able to reliably observe down to 1 to 2 ppb at the surface."

Page 8, line 24: it appears that a uniform profile is assumed here. Is this really a good assumption?

Good point, and one we should have highlighted in the original manuscript. This is a simplifying assumption, which we have made for three key reasons: 1) We believe that given the level of uncertainty inherent in our box model (with its assumptions of no transport and the use of a range of plausible effective lifetimes of NO2 and NH3), it is preferable to keep the model as simple as possible to both make it as transparent as possible, and to avoid giving the impression of a level of accuracy or certainty in modeled emissions that is not present, especially given the uncertainty in vertical distribution of NH3 in the IASI retrievals. 2) Independent of point 1, we do not think the assumption is unreasonable. The vertical profile of IASI NH3 is assumed to be essentially homogeneous within the boundary layer. And since lifetimes of both gases are relatively short, it seems plausible that gases are concentrated near the surface, especially when concentrations are most influenced by soil fluxes. 3) Lastly, we decided to take an identical modeling approach to that of Whitburn et al 2015 in modeling NH3 fluxes so that the results of the

two studies, which focus on adjoining regions in Africa and contrasting sources of NH3, are strictly comparable.

In addition to emphasizing the simple nature of the box model in the text, we have added the following clarifying text (Page 17, line 26):

"It is also important to note the simplifying assumption of a uniform atmospheric profile in the box model, which ignores any variation in vertical distribution. Still, we believe a uniform profile is a reasonable assumption, especially for soil fluxes. Each of these gases has relatively short lifetimes, and unlike fire plumes, are unlikely to be rapidly lofted to high altitudes following emission from soil. In addition, there is very little variation in NH3 distribution throughout the boundary layer in the assumed IASI profile."

*Whitburn et al. 2015 Atmos. Env. 121: 42e54*

Page 11, line 11: I don't see the increase in fire emissions in late March

Thank you for pointing out this error; the modest increase is actually just prior to the April pulse, not the March pulse. We have revised the paragraph to read as follows (Page 11, line 16; changes in bold italics):

"For our focal region of the Sahel (defined above and outlined in red in Figure 1), mean atmospheric $NH_3$ concentrations exhibit two distinct peaks in late March and April (Figure 2a, highlighted in light and dark pink, respectively), which represent the highest concentrations observed in 2008. ***The late March peak occurs at the same time as an apparent modest increase in mean soil moisture (Fig 2b).*** The peak in April, during which atmospheric $NH_3$ concentrations over the Sahel are elevated relative to other parts of north equatorial Africa (Figure 3a), occurs during the first period of sustained rainfall in the focal region, and corresponds to a peak in soil moisture, suggesting a possible causal relationship between changes in soil moisture and atmospheric $NH_3$ concentrations (Figure 2b and section 3.2.1 below). ***This April peak does occur following a possible modest increase mean fire emissions (Fig 2a) across the Sahel.*** Overall, however, the seasonality in IASI-retrieved atmospheric $NH_3$ concentrations exhibits a marked difference from the seasonality in GFED4s $NH_3$ emissions from fires, which start increasing in September and peak in November (Fig 2a and section 3.2.2 below)."

Page 12, line 25: It could be helpful to rewrite this as: " except during the biomass burning season"

We have revised the sentence to read as follows (Page 13, line 8; added text in bold italics):

"A comparison between our simple box model estimates of $NH_3$ flux and emissions from the GFED4s inventory strongly supports the hypothesis that biomass burning does not play an important role in $NH_3$ emissions during March or April, and further suggests that biomass

burning may represent a relatively unimportant regional source of NH$_3$ during most of the year, *outside of the biomass burning season* (Figure 6)."

Page 13, line 23: at what scale where these correlations calculated: 0.25 deg or over the entire box?

 The correlations were calculated over the entire box; we have revised the sentence to clarify (Page 14, line 10; new text in bold italics):

"For the month of April, total column NH$_3$ concentrations and tropospheric NO$_2$ concentrations *integrated across the entire focal region* are strongly correlated (r=0.78, p<0.0001)."

Page 15, line 22: possibly repeat that IASI NH3 appears to be biased low

Thank you for the suggestion. We have revised the paragraph to read as follows (Page 17, line 16; new text in bold italics):

"During the early growing season, atmospheric concentrations of NH$_3$ are roughly an order of magnitude higher than NO$_2$ (figure 2). It is important to note that NH$_3$ retrievals generally have a higher error, and that our screening process may introduce a potential bias in that we permit retrievals with higher uncertainty if they are low concentrations; we also retain observations of negative concentrations. *In addition, as mentioned earlier, compared to FTIR observations the IASI total columns are biased low by ~30% which varies per region depending on the local concentrations*. From this perspective, our concentration and emission estimates can be considered conservative."

Page 17, line 17: only NO2 concentrations become comparable to NH3, not the fluxes

 Well noted—as it turns out, neither NO2 concentrations nor fluxes are comparable in magnitude to those of NH3 (note the differences in scale in figure 2, something we clearly failed to do adequately), so thanks very much for catching this mis-statement. We have revised the sentence to read as follows (Page 20, line 14; new text in bold italics, including new text to clarify the time period being discussed):

"After the early season pulses, NH$_3$ concentrations and fluxes remain fairly elevated (figure 2, 6); during this period *of May through July*, NO$_2$ concentrations and fluxes increase, becoming *closer* in magnitude to NH$_3$ (figure 2,6)."

Page 18, line 8: a more sophisticated inverse modeling of the emissions? This should be made clearer, and should suggest an approach

This point is really just intended to acknowledge that our emissions modeling is extremely simple, and that, for example, inverse modeling using a Kalman filtering or Bayesian approach and an atmospheric chemical transport model could potentially provide more insight into surface

emissions. We add the following text to provide an illustration and example of what is intended with this statement (Page 21, line 4; new text in bold italics):

"A more sophisticated inverse modelling approach *(e.g., using a Bayesian approach in combination with an atmospheric chemical transport model)* could provide firmer insight into the magnitude of emissions, as well as provide some insight into the magnitude of specific NH$_3$ sources."

Page 18, line 10: the increased cloud cover during the rainy season could certainly mask increased NH3. Cloud cover would also impact OMI NO2, but possibly not to the same degree. If at all possible, the authors should discuss this.

A good point. In the original manuscript, for the correlation analysis between satellite NO2 and NH3 observations, only grid cells including observations of both gases were used in the analysis, and so no effect of cloud cover should be present in that analysis. In contrast, figures 2 and 6 (which present both NO2 and NH3 data), are based on OMI-NO2 observations only from grid cells that also had IASI-NH3 observations. The reasoning for not also screening NH3 observations based on whether NO2 observations were present was that the focus of the manuscript is NH3, and the NO2 data were intended as a complementary measure that could provide insight into the mechanism behind the NH3 pulses. We agree that clearly illustrating and discussing the mismatch between NH3 and NO2 responses to wetting events is important, and so in revision, we include additional analyses restricted to grid cells that have both IASI and OMI observations, resulting in new versions of figures 2 and 6 (included below as Figures S1 and S3, respectively); these figures suggest that the different seasonality of the two gases is not an artifact of different impacts of cloud cover on the two satellite products. We have decided that in the main text of the manuscript it is more appropriate to present NH3 that is not screened based on the presence of NO2 observations, and to include the new versions in the supplemental information: we feel that the goal of presenting as complete a picture of the seasonality of NH3 concentrations as possible takes precedence over the goal of comparing NH3 and NO2 patterns. We do refer to the two new figures at Page 15 line 26 and Page 16 line 20:

> "for a strict comparison in which concentrations of NO2 and NH3 are calculated using only grid cells that have observations for both gases, see Figure S1, though the results are qualitatively similar to Figure 2"

> "for a presentation of modelled emissions calculated based only grid cells that have satellite observations for both gases, see Figure S3. This screening reduces annual modelled NH$_3$ emissions by roughly 5%, and reduces modelled emissions during the pulses by roughly 2-3% relative to modelled NH$_3$ emissions that do not take the presence or absence of NO$_2$ observations into account."

When including only grid cells with both NH3 and NO2 observations, there is little qualitative change in seasonal patterns. In the revised figure 6 (Figure S3), we also use screened GFED NO2 and NH3 data, so that only observations from pixels where the respective satellite observations also exist are included in calculating the daily means presented to allow for a more direct comparison to modelled fluxes; again, there are no qualitative changes in the patterns observed. We also did a small quantitative check on the effect on modelled emissions of screening NH3 based on NO2 observations, and found that doing so results in a decline in total annual modelled emissions of

roughly 5%, and a decline in emissions during the pulses of roughly 2 to 3%, making the pulses slightly smaller but slightly more important in the context of annual emissions.

In addition, in re-visiting our screening process, we realized that we had used the wrong mask to prepare the NO2 data for the box model and figure 6. We have since done a thorough check to ensure that the masks and analyses conducted for the revision were completed as intended. The change in mask does not affect the results qualitatively; there is, however, a substantial increase in the magnitude of modelled NO2 emissions. Only figure 6 is affected by the change (the correct mask was used in preparing figure 2). We have revised figure 6, and deleted the phrase "possibly by a factor of 10 or more," originally on page 15 line 27.

[Figure]

Figure S3. *Comparison of daily total surface fluxes and biomass burning emissions of a) NH₃-N and b) NO₂-N for the focal study region in the Sahel during 2008.* Total surface fluxes are estimated from IASI NH₃ and OMI NO₂ observations using a simple box model and assuming effective lifetimes of 12, 24, or 36 hours for NH₃ and of 6, 12, or 24 hours for NO₂. Fire emissions are taken from the GFED4s database. Modelled and GFED4s mean emissions were calculated using values only of 0.25° grid cells within the study region for which both NO₂ and NH₃ observations were present. Note the difference in scales, and that shorter effective lifetimes result in higher modelled emissions.

[Figure]

Figure S1: *Early growing season NH₃ pulses temporally associated with changes in soil moisture and with NO₂ pulses over the focal study region in the Sahel during 2008.* Top panel: Daily atmospheric NH₃ concentrations from IASI and NH₃ biomass burning emissions from GFED4s. Middle panel: ESA-CCI soil moisture and TRMM precipitation. Bottom panel: atmospheric NO₂ concentrations from OMI. Putative soil emission pulses in March and April are highlighted in bright and dark pink, respectively. The means of all variables presented were calculated using values of 0.25° grid cells within the study region for which both NO₂ and NH₃ observations were present. GFED4s emissions were converted from a mass-based to molecule-based flux to allow comparison with the IASI retrievals; note the different scales for the left and right y axes of the top panel.

Minor comments:

Page 4, line 27: the second sentence does not follow from the first

We presume the issue being raised here is the use of the East African example and the leap from a discussion of environmental conditions to examples of ammonia emissions being structurally inelegant (a point we agree with). We have re-written the paragraph as follows (page 4 line 15; moved or added text in bold italics):

"Given the importance of rainfall seasonality, soil pH, and N availability in contributing to $NH_3$ emission pulses, soils in the Sahel may be an important source of $NH_3$ to the atmosphere during the onset of the rainy season*, and a case study for determining whether Birch effect $NH_3$ pulsing is an important process at broad regional scales*. The Sahel is a grassland environment representing a transition between desert and productive savannas. It is characterized by a unimodal rainfall seasonality, with mean annual precipitation typically ranging between 100 and 600 mm $yr^{-1}$. Seasonal variation in rainfall is broadly determined by movement of the Intertropical Convergence Zone (ITCZ). Migration of the ITCZ north of the equator in the first half of the calendar year is accompanied by the onset of the rainy season and West African Monsoon, with the first substantial rain events occurring in April. The southward retreat of the ITCZ marks the dry season *in the Sahel* starting in October or November. Recent maps of African soils based on surface reflectance suggest that soils across the Sahel tend to have pHs largely near neutral, but can be higher than 9 in some areas (Vågen et al., 2016). The combination of seasonal rainfall variability and soils with neutral or alkaline pHs suggests that Sahelian soils may be an important source of $NH_3$ at the onset of the rainy season. Although the Sahel has regions of relatively dense cropland, it is characterized by lower levels of fertilizer inputs (FAO, accessed 2018) and smaller loads of atmospheric N deposition (Dentener et al., 2006; Galy-Lacaux and Delon, 2014; Laouali et al., 2012; though deposition can be elevated at the Sahel's southern boundary) than other parts of the world. However, it has moderately high livestock densities (Robinson et al., 2014), potentially providing sites of abundant available N for the production of $NH_3$. *Indeed, soil $NH_3$ emissions have been shown to be higher at a site in northern Senegal following a rain event (Delon et al., 2017).*"

Page 7, line 7: IASI data are only used if the pixels are 75% cloud-free, while OMI data is included only if the cloud-cover is less than 30%. While these statements are almost equivalent, it would be clearer to say 75% cloud-free and 70% cloud-free, or cloud cover less than 25% and cloud cover less than 30%.

We have changed the description of the OMI level 3 NO2 product as follows (page 8 line 10):

"The product is cloud-screened, including only pixels that are at least 70% cloud-free"

[revised manuscript text omitted]